# Justification of Direct Scheme for Asymptotic Solving Three-Tempo Linear-Quadratic Control Problems under Weak Nonlinear Perturbations

**Galina Kurina [1,2,*] and Margarita Kalashnikova [3]**

1   Faculty of Mathematics, Voronezh State University, Universitetskaya pl., 1, 394018 Voronezh , Russia
2   Federal Research Center "Computer Science and Control" of RAS, ul. Vavilova, 44/2, 119333 Moscow, Russia
3   Atos IT Solutions and Services, pr. Truda, 65, 394026 Voronezh, Russia
*   Correspondence: kurina@math.vsu.ru

**Abstract:** The paper deals with an application of the direct scheme method, consisting of immediately substituting a postulated asymptotic solution into a problem condition and determining a series of control problems for finding asymptotics terms, for asymptotics construction of a solution of a weakly nonlinearly perturbed linear-quadratic optimal control problem with three-tempo state variables. For the first time, explicit formulas for linear-quadratic optimal control problems, from which all terms of the asymptotic expansion are found, are justified, and the estimates of the proximity between the asymptotic and exact solutions are proved for the control, state trajectory, and minimized functional. Non-increasing of the minimized functional, if a next approximation to the optimal control is used, following from the proposed algorithm of the asymptotics construction, is also established.

**Keywords:** optimal control problems; weak nonlinear perturbations; three-tempo variables; asymptotic solutions; the direct scheme method; estimates of asymptotic solution

## 1. Introduction

Systems with two-tempo variables are the main object in the study of singularly perturbed control problems (see, for instance, the reviews [1–3]). However, many practical problems contain multi-tempo fast variables. For instance, such variables arise in models of chain chemical reactions [4], fuel cells with a proton membrane [5], electrical chains [6], electromechanical processes in a synchronous machine [7], power systems [8], nuclear reactors [9], aircraft [10], ocean currents [11], rolling mills [12], two-wheeled carriages [13], forest pests [14], and epidemics [15].

Various asymptotic and numerical (see, for instance, [16]) methods are used for studying singularly perturbed systems with many small parameters standing before derivatives. Basic methods of asymptotic analysis are boundary functions method [17] and integral manifolds method ([18], ch. 7–10), which reduce the considered problem to a problem of simpler structure. The limit passage of an initial problem solution of a system with many small parameters at derivatives, when these parameters tend to zero, was studied for the first time by A.N. Tikhonov [19] and I.S. Gradstein [20]. Asymptotic solution of such problems was first constructed by A.B. Vasil'eva [21].

There are two approaches to constructing asymptotic solutions of optimal control problems. The traditional one is based on an asymptotic solution of a system following from control optimality conditions. Another approach, called the direct scheme method, consists of immediately substituting a postulated asymptotic expansion of a solution into the problem condition and receiving a series of problems for finding asymptotic terms. For two-tempo systems, it is presented, for example, in [22,23]. This method allows for

establishing non-increasing of values of the minimized functional if a next optimal control approximation is used. Moreover, standard programs for solving optimal control problems can be applied for finding asymptotics terms. The direct scheme method has been, for instance, used in [24] to obtain any order asymptotic solution of a linear-quadratic optimal control problem with cheap controls of different costs.

The present paper deals with an asymptotic solution construction for the problem $P_\varepsilon$ with weak nonlinear perturbations in a quadratic performance index and in a linear state equation. Namely, the following functional

$$J_\varepsilon(u) = \int_0^T \left(1/2(w(t,\varepsilon)'W(t)w(t,\varepsilon) + u(t,\varepsilon)'R(t)u(t,\varepsilon)) + \varepsilon F(w(t,\varepsilon), u(t,\varepsilon), t, \varepsilon)\right) dt \quad (1)$$

is minimized on trajectories of three-tempo singularly perturbed system

$$\mathcal{E}(\varepsilon)\frac{dw(t,\varepsilon)}{dt} = A(t)w(t,\varepsilon) + B(t)u(t,\varepsilon) + \varepsilon f(w(t,\varepsilon), u(t,\varepsilon), t, \varepsilon), \ t \in [0,T], \quad (2)$$

with the initial condition

$$w(0,\varepsilon) = w^0. \quad (3)$$

Here, $\varepsilon$ is a non-negative small parameter, $T > 0$ is fixed, the prime means transposition; $w(t,\varepsilon) = (x(t,\varepsilon)', y(t,\varepsilon)', z(t,\varepsilon)')'$, $x(t,\varepsilon) \in \mathbb{R}^{n_1}$, $y(t,\varepsilon) \in \mathbb{R}^{n_2}$, $z(t,\varepsilon) \in \mathbb{R}^{n_3}$, $u(t,\varepsilon) \in \mathbb{R}^m$; $\mathcal{E}(\varepsilon) = diag(I_{n_1}, \varepsilon I_{n_2}, \varepsilon^2 I_{n_3})$, $I_{n_i}$ is the identity matrix of order $n_i$, $f = (f^{(1)\prime}, f^{(2)\prime}, f^{(3)\prime})'$, $f^{(i)} \in \mathbb{R}^{n_i}$, $B = (B^{(1)\prime}, B^{(2)\prime}, B^{(3)\prime})'$, $B^{(i)} : \mathbb{R}^m \to \mathbb{R}^{n_i}$, $i = \overline{1,3}$; all functions in (1), (2) are sufficiently smooth with respect to their arguments; for all $t \in [0,T]$ matrices $W(t)$, $R(t)$ are symmetric, moreover, $W(t)$, $R(t)$ and $S(t) = B(t)R(t)^{-1}B(t)'$ are positive definite.

It is assumed that the stability of the matrices $A_{33}$ and $A_{22} - A_{23}A_{33}^{-1}A_{32}$ takes place. Here, and further $A_{ij}$, $i, j = \overline{1,3}$, mean matrices from a block representation of a matrix $A$ with number of rows and columns $n_1$, $n_2$, $n_3$.

In contrast to [25], where optimal control problems for finding some zero order asymptotics terms for a solution of a nonlinear singularly perturbed problem with three-tempo state variables were formulated, here, explicit expressions of problems for receiving all asymptotic terms are obtained. Note that explicit formulas are very useful for research applying asymptotic methods for solving practical problems.

It should be noted that some results concerning the algorithm of asymptotic solving problem (1)–(3) have been presented in [26], but rigorous proofs and estimates are absent there. Note that [26] deal with matrices in (1), (2) depending on $\varepsilon$. However, expanding these matrices with respect to non-negative integer powers of $\varepsilon$ and including terms depending on $\varepsilon$ into the small nonlinearities, we obtain the problem $P_\varepsilon$ in our statement.

It is well known that, if a linear-quadratic problem is nonsingular, then its solving is reduced to solving a system of linear differential equations resolved with respect to derivatives. Under studying nonlinear singularly perturbed optimal control problems, it is ordinarily assumed that the control problem is nonsingular, i.e., an optimal control is presented as an explicit function with respect to state and costate variables. See e.g., [27], where, apparently for the first time, singular perturbations methods were used for optimal control problems. In the present paper, unlike these cases, we do not assume the non-singularity of the considered problem for all $\varepsilon$ and, for obtaining asymptotic estimates, we analyze a nonlinear singularly perturbed differential-algebraic system.

The essential new results obtained in this paper for problem (1)–(3) are the following:

1. The rigorous justification of explicit forms of linear-quadratic optimal control problems, solutions of which are used under constructing an asymptotic solution of nonlinear problem (1)–(3);

2. The proof of estimates of the proximity between the exact solution and asymptotic one obtained by the direct scheme method for the control, state trajectory of system (2), (3), and functional (1);

3. The proof of non-increasing values of functional (1) under using new asymptotic approximations to the optimal control and constructing minimized sequences.

Throughout the paper, the coefficient with $\varepsilon^i$ in an expansion of a function $\omega = \omega(\varepsilon)$ in a series in powers of $\varepsilon$ will be denoted by $\omega_i$ or $[\omega]_i$. The $k$-th partial sum of a series will be denoted by upper wave and the low index $k$ or by braces with the low index k, i.e., $\widetilde{\omega}_k = \{\omega\}_k = \sum_{j=0}^{k} \varepsilon^j \omega_j$. The functions with negative indices will be considered equal to zero. Positive constants in estimates will be denoted as $c$ and $\text{æ}$.

The paper is organized as follows: in Section 2, we present a formalism of asymptotics construction. Optimal control problems for finding asymptotic terms are given in Section 3. Section 4 is devoted to justification of such a choice of control problems. Namely, transformations of coefficients of expansion of minimized functional with respect to powers of $\varepsilon$ with even and odd indices are considered. Asymptotic estimates of the proximity between the asymptotic and exact solutions are proved in Section 5. Non-increasing of the minimized functional, if a next optimal control approximation is used, is also discussed in this section. The last Section 6 contains conclusions.

## 2. Formalism of Asymptotics Construction

Following the boundary function method by A.B. Vasil'eva (see, for instance, [28]), we will seek a solution of problem (1)–(3) in the form

$$\vartheta(t, \varepsilon) = \overline{\vartheta}(t, \varepsilon) + \sum_{i=0}^{1} (\Pi_i \vartheta(\tau_i, \varepsilon) + Q_i \vartheta(\sigma_i, \varepsilon)). \tag{4}$$

Here, $\vartheta(t, \varepsilon) = (w(t, \varepsilon)', u(t, \varepsilon)')'$, $\overline{\vartheta}(t, \varepsilon) = \sum_{j \geq 0} \varepsilon^j \overline{\vartheta}_j(t)$, $\Pi_i \vartheta(\tau_i, \varepsilon) = \sum_{j \geq 0} \varepsilon^j \Pi_{ij} \vartheta(\tau_i)$, $Q_i \vartheta(\sigma_i, \varepsilon) = \sum_{j \geq 0} \varepsilon^j Q_{ij} \vartheta(\sigma_i)$, $\tau_i = t/\varepsilon^{i+1}$, $\sigma_i = (t - T)/\varepsilon^{i+1}$, $i = 0, 1$, $\overline{\vartheta}_j(t)$ are regular functions, $\Pi_{ij} \vartheta(\tau_i)$ and $Q_{ij} \vartheta(\sigma_i)$ are boundary functions of exponential type in neighborhoods $t = 0$ and $t = T$, respectively, i.e.,

$$\|\Pi_{ij} \vartheta(\tau_i)\| \leqslant c \exp(-\text{æ}\tau_i), \ \tau_i \geqslant 0, \ \|Q_{ij} \vartheta(\sigma_i)\| \leqslant c \exp(\text{æ}\sigma_i), \ \sigma_i \leqslant 0,$$

where $c$ and $\text{æ}$ are positive constants independent of the arguments of functions under study.

For any sufficiently smooth function $G(w(t, \varepsilon), u(t, \varepsilon), t, \varepsilon)$, we will use the notation $G(\vartheta(t, \varepsilon), t, \varepsilon)$ and the asymptotic representation

$$G(\vartheta, t, \varepsilon) = \overline{G}(t, \varepsilon) + \sum_{i=0}^{1} (\Pi_i G(\tau_i, \varepsilon) + Q_i G(\sigma_i, \varepsilon)), \tag{5}$$

$\overline{G}(t, \varepsilon) = G(\overline{\vartheta}(t, \varepsilon), t, \varepsilon) = \sum_{j \geq 0} \varepsilon^j \overline{G}_j(t)$, $\Pi_0 G(\tau_0, \varepsilon) = G(\overline{\vartheta}(\varepsilon\tau_0, \varepsilon) + \Pi_0 \vartheta(\tau_0, \varepsilon), \varepsilon\tau_0, \varepsilon)$ $- G(\overline{\vartheta}(\varepsilon\tau_0, \varepsilon), \varepsilon\tau_0, \varepsilon) = \sum_{j \geq 0} \varepsilon^j \Pi_{0j} G(\tau_0)$, $\Pi_1 G(\tau_1, \varepsilon) = G(\overline{\vartheta}(\varepsilon^2\tau_1, \varepsilon) + \Pi_0 \vartheta(\varepsilon\tau_1, \varepsilon)$ $+ \Pi_1 \vartheta(\tau_1, \varepsilon), \varepsilon^2\tau_1, \varepsilon) - G(\overline{\vartheta}(\varepsilon^2\tau_1, \varepsilon) + \Pi_0 \vartheta(\varepsilon\tau_1, \varepsilon), \varepsilon^2\tau_1, \varepsilon) = \sum_{j \geq 0} \varepsilon^j \Pi_{1j} G(\tau_1)$, $Q_0 G(\sigma_0, \varepsilon) = G(\overline{\vartheta}(T + \varepsilon\sigma_0, \varepsilon) + Q_0 \vartheta(\sigma_0, \varepsilon), T + \varepsilon\sigma_0, \varepsilon) - G(\overline{\vartheta}(T + \varepsilon\sigma_0, \varepsilon), T + \varepsilon\sigma_0, \varepsilon)$ $= \sum_{j \geq 0} \varepsilon^j Q_{0j} G(\sigma_0)$, $Q_1 G(\sigma_1, \varepsilon) = G(\overline{\vartheta}(T + \varepsilon^2\sigma_1, \varepsilon) + Q_0 \vartheta(\varepsilon\sigma_1, \varepsilon) + Q_1 \vartheta(\sigma_1, \varepsilon), T + \varepsilon^2\sigma_1, \varepsilon)$ $- G(\overline{\vartheta}(T + \varepsilon^2\sigma_1, \varepsilon) + Q_0 \vartheta(\varepsilon\sigma_1, \varepsilon), T + \varepsilon^2\sigma_1, \varepsilon) = \sum_{j \geq 0} \varepsilon^j Q_{1j} G(\sigma_1)$.

Substitute (4) in (1) and present the integrand in the form of sum (4). Passing in the integrals from the expressions depending on $\tau_i$, $\sigma_i$, $i = 0, 1$, to integrals over the corresponding intervals $[0, +\infty)$ and $(-\infty, 0]$, we obtain the following expansion of the functional (1)

$$J_\varepsilon(u) = \sum_{j \geq 0} \varepsilon^j J_j. \tag{6}$$

Substituting expansion (4) into system (2) and initial value (3), using (5), then equating terms of the same powers of $\varepsilon$, separately depending on regular and different boundary functions, we obtain relations for defining asymptotics terms.

Introducing the notation $E_1 = diag(I_{n_1}, 0, 0)$, $E_2 = diag(0, I_{n_2}, 0)$, $E_3 = diag(0, 0, I_{n_3})$, and $\phi(\vartheta, t, \varepsilon) = A(t)w(t, \varepsilon) + B(t)u(t, \varepsilon) + \varepsilon f(w(t, \varepsilon), u(t, \varepsilon), t, \varepsilon)$, we obtain the following equations:

$$E_1 \frac{d\overline{w}_j(t)}{dt} + E_2 \frac{d\overline{w}_{j-1}(t)}{dt} + E_3 \frac{d\overline{w}_{j-2}(t)}{dt} = [\overline{\phi}(t, \varepsilon)]_j, \tag{7}$$

$$(E_1 + E_2) \frac{d\Pi_{0j}w(\tau_0)}{d\tau_0} + E_3 \frac{d\Pi_{0(j-1)}w(\tau_0)}{d\tau_0} = E_1 [\Pi_0 \phi(\tau_0, \varepsilon)]_{j-1} \\ + (E_2 + E_3)[\Pi_0 \phi(\tau_0, \varepsilon)]_j, \tag{8}$$

$$(E_1 + E_2) \frac{dQ_{0j}w(\sigma_0)}{d\sigma_0} + E_3 \frac{dQ_{0(j-1)}w(\sigma_0)}{d\sigma_0} = E_1 [Q_0 \phi(\sigma_0, \varepsilon)]_{j-1} \\ + (E_2 + E_3)[Q_0 \phi(\sigma_0, \varepsilon)]_j, \tag{9}$$

$$\frac{d\Pi_{1j}w(\tau_1)}{d\tau_1} = E_1 [\Pi_1 \phi(\tau_1, \varepsilon)]_{j-2} + E_2 [\Pi_1 \phi(\tau_1, \varepsilon)]_{j-1} + E_3 [\Pi_1 \phi(\tau_1, \varepsilon)]_j, \tag{10}$$

$$\frac{dQ_{1j}w(\sigma_1)}{d\sigma_1} = E_1 [Q_1 \phi(\sigma_1, \varepsilon)]_{j-2} + E_2 [Q_1 \phi(\sigma_1, \varepsilon)]_{j-1} + E_3 [Q_1 \phi(\sigma_1, \varepsilon)]_j. \tag{11}$$

From Equations (8)–(11) at $j = 0$, (10) and (11) at $j = 1$, we found the corresponding boundary functions

$$E_1 \Pi_{00}w(\tau_0) = 0, \ E_1 \Pi_{10}w(\tau_1) = E_1 \Pi_{11}w(\tau_1) = 0, \ E_1 Q_{00}w(\sigma_0) = 0, \\ E_1 Q_{10}w(\sigma_1) = E_1 Q_{11}w(\sigma_1) = 0, \ E_2 \Pi_{10}w(\tau_1) = 0, \ E_2 Q_{10}w(\sigma_1) = 0. \tag{12}$$

In view of the last equalities, from (3), we obtain relations for initial values

$$E_1 \overline{w}_0(0) = E_1 w^0, \ E_1(\overline{w}_1(0) + \Pi_{01}w(0)) = 0, \tag{13}$$

$$E_1(\overline{w}_j(0) + \Pi_{0j}w(0) + \Pi_{1j}w(0)) = 0, \ j \geq 2, \tag{14}$$

$$E_2(\overline{w}_0(0) + \Pi_{00}w(0)) = E_2 w^0, \tag{15}$$

$$E_2(\overline{w}_j(0) + \Pi_{0j}w(0) + \Pi_{1j}w(0)) = 0, \ j \geq 1, \tag{16}$$

$$E_3(\overline{w}_j(0) + \Pi_{0j}w(0) + \Pi_{1j}w(0)) = \begin{cases} E_3 w^0, \ j = 0, \\ 0, \ j \geq 1. \end{cases} \tag{17}$$

**Remark 1.** *If boundary functions $\Pi_{ij}w$, $Q_{ij}w$, $i = 0, 1$, $j = \overline{0, n-1}$ have been found, then, from Equations (8)–(11), it follows the corollary that functions $E_1 \Pi_{in}w(\tau_i)$, $E_1 Q_{in}w(\sigma_i)$, $i = 0, 1$, $E_2 \Pi_{1n}w(\tau_1)$, $E_2 Q_{1n}w(\sigma_1)$, and $E_1 \Pi_{1(n+1)}w(\tau_1)$, $E_1 Q_{1(n+1)}w(\sigma_1)$ are known.*

## 3. Optimal Control Problems for Finding Asymptotics Terms

In this section, forms of control problems for finding asymptotics terms will be given. In contrast to [26], the justification of these relations will be presented.

With the help of the notations,

$$\rho(\vartheta, \psi, t, \varepsilon) = W(t)w(t, \varepsilon) - A(t)'\psi(t, \varepsilon) + \varepsilon(F_w(\vartheta, t, \varepsilon)' - f_w(\vartheta, t, \varepsilon)'\psi(t, \varepsilon)), \\ \chi(\vartheta, \psi, t, \varepsilon) = R(t)u(t, \varepsilon) - B(t)'\psi(t, \varepsilon) + \varepsilon(F_u(\vartheta, t, \varepsilon)' - f_u(\vartheta, t, \varepsilon)'\psi(t, \varepsilon)),$$

five optimal control problems $\overline{P}_j$, $\Pi_{ij}P$, $Q_{ij}P$, $i = 0, 1$, for determining asymptotics terms in expansion (4) will be written. Costate variables in these problems will be denoted as $\overline{\psi}_j(t)$, $\Pi_{ij}\psi(\tau_i)$, $Q_{ij}\psi(\sigma_i)$, $i = 0, 1$, respectively.

Furthermore, the hat and the low index $k$ in a function notation will mean that the function is calculated with the functional argument equal to the $k$-th partial sum of the corresponding expansion, e.g., $\widehat{\bar{f}}_k(t,\varepsilon) = f(\widetilde{\bar{\vartheta}}_k(t,\varepsilon),t,\varepsilon)$.

In the following expressions with $\rho$ and $\chi$ in the performance indices of the formulated optimal control problems, we take $\psi(t,\varepsilon) = \sum_{j=0}^{\infty} \varepsilon^j (\bar{\psi}_j(t) + (\varepsilon E_1 + E_2 + E_3)(\Pi_{0j}\psi(\tau_0) + Q_{0j}\psi(\sigma_0)) + (\varepsilon^2 E_1 + \varepsilon E_2 + E_3)(\Pi_{1j}\psi(\tau_1) + Q_{1j}\psi(\sigma_1)))$.

Regular functions $\bar{\vartheta}_j(t)$, $t \in [0,T]$, are determined as solutions of problems $\bar{P}_j$, which consist of minimizing the functional

$$\bar{J}_j(\bar{u}_j) = \bar{w}_j(T)' E_1(Q_{0(j-1)}\psi(0) + Q_{1(j-2)}\psi(0)) + \int_0^T (\bar{w}_j(t)'(\frac{1}{2}W(t)\bar{w}_j(t)$$

$$+[\widehat{\bar{\rho}}_{j-1}(t,\varepsilon)]_j - E_2\frac{d\bar{\psi}_{j-1}(t)}{dt} - E_3\frac{d\bar{\psi}_{j-2}(t)}{dt}) + \bar{u}_j(t)'(\frac{1}{2}R(t)\bar{u}_j(t) + [\widehat{\bar{\chi}}_{j-1}(t,\varepsilon)]_j)) \, dt$$

on trajectories of system (7) with initial conditions from (13) or (14) in dependence on $j$.

The boundary functions $\Pi_{0j}\vartheta(\tau_0)$, $\tau_0 \in [0,+\infty)$ are determined from optimal control problems $\Pi_{0j}P$ consisting of minimizing the functional

$$\Pi_{0j}J(\Pi_{0j}u) = \int_0^{+\infty} (\Pi_{0j}w(\tau_0)'(\frac{1}{2}W(0)\Pi_{0j}w(\tau_0) + [\widehat{\Pi}_{0(j-1)}\rho(\tau_0,\varepsilon)]_j - E_3\frac{d\Pi_{0(j-1)}\psi(\tau_0)}{d\tau_0})$$

$$+\Pi_{0j}u(\tau_0)'(\frac{1}{2}R(0)\Pi_{0j}u(\tau_0) + [\widehat{\Pi}_{0(j-1)}\chi(\tau_0,\varepsilon)]_j)) \, d\tau_0$$

on trajectories of system (8) with the conditions $\Pi_{0j}x(+\infty) = 0$ and (15) or (16) in dependence on $j$.

The boundary functions $Q_{0j}\vartheta(\sigma_0)$, $\sigma_0 \in (-\infty, 0]$, are determined from optimal control problems $Q_{0j}P$ consisting of minimizing the functional

$$Q_{0j}J(Q_{0j}u) = Q_{0j}w(0)' E_2(\bar{\psi}_j(T) + Q_{1(j-1)}\psi(0))$$

$$+ \int_{-\infty}^0 (Q_{0j}w(\sigma_0)'(\frac{1}{2}W(T)Q_{0j}w(\sigma_0) + [\widehat{Q}_{0(j-1)}\rho(\sigma_0,\varepsilon)]_j - E_3\frac{dQ_{0(j-1)}\psi(\sigma_0)}{d\sigma_0})$$

$$+Q_{0j}u(\sigma_0)'(\frac{1}{2}R(T)Q_{0j}u(\sigma_0) + [\widehat{Q}_{0(j-1)}\chi(\sigma_0,\varepsilon)]_j)) \, d\sigma_0$$

on trajectories of system (9) with the condition $(E_1 + E_2)Q_{0j}w(-\infty) = 0$.

The boundary functions $\Pi_{1j}\vartheta(\tau_1)$, $\tau_1 \in [0,+\infty)$, are determined from optimal control problems $\Pi_{1j}P$ consisting of minimizing the functional

$$\Pi_{1j}J(\Pi_{1j}u) = \int_0^{+\infty} (\Pi_{1j}w(\tau_1)'(\frac{1}{2}W(0)\Pi_{1j}w(\tau_1) + [\widehat{\Pi}_{1(j-1)}\rho(\tau_1,\varepsilon)]_j)$$

$$+\Pi_{1j}u(\tau_1)'(\frac{1}{2}R(0)\Pi_{1j}u(\tau_1) + [\widehat{\Pi}_{1(j-1)}\chi(\tau_1,\varepsilon)]_j)) \, d\tau_1$$

on trajectories of system (10) with the conditions $(E_1 + E_2)\Pi_{1j}w(+\infty) = 0$ and (17).

The boundary functions $Q_{1j}\vartheta(\sigma_1)$, $\sigma_1 \in (-\infty, 0]$, are determined from optimal control problems $Q_{1j}P$ consisting of minimizing the functional

$$Q_{1j}J(Q_{1j}u) = Q_{1j}w(0)' E_3(\bar{\psi}_j(T) + Q_{0j}\psi(0)) + \int_{-\infty}^0 (Q_{1j}w(\sigma_1)'(\frac{1}{2}W(T)Q_{1j}w(\sigma_1)$$

$$+[\widehat{Q}_{1(j-1)}\rho(\sigma_1,\varepsilon)]_j) + Q_{1j}u(\sigma_1)'(\frac{1}{2}R(T)Q_{1j}u(\sigma_1) + [\widehat{Q}_{1(j-1)}\chi(\sigma_1,\varepsilon)]_j)) \, d\sigma_1$$

on trajectories of system (11) with the condition $Q_{1j}w(-\infty) = 0$.

**Remark 2.** *Though the original problem* (1)–(3) *is nonlinear, the considered optimal control problems* $\overline{P}_j$, $\Pi_{ij}P$, $Q_{ij}P$, $i = 0, 1$, *are linear-quadratic.*

Solutions of the formulated optimal control problems can be found from the control optimality conditions in the Pontryagin maximum principle form. Namely, a solution of the problem $\overline{P}_j$ can be found from (7), (13), or (14) in dependence on $j$, and the relations

$$B(t)'\overline{\psi}_j(t) - R(t)\overline{u}_j(t) - [\widehat{\overline{\chi}}_{j-1}(t,\varepsilon)]_j = 0, \tag{18}$$

$$E_1\frac{d\overline{\psi}_j(t)}{dt} = W(t)\overline{w}_j(t) - A(t)'\overline{\psi}_j(t) + [\widehat{\overline{\rho}}_{j-1}(t,\varepsilon)]_j - E_2\frac{d\overline{\psi}_{j-1}(t)}{dt} - E_3\frac{d\overline{\psi}_{j-2}(t)}{dt}, \tag{19}$$

$$E_1\overline{\psi}_j(T) = -E_1(Q_{0(j-1)}\psi(0) + Q_{1(j-2)}\psi(0)). \tag{20}$$

A solution of the problem $\Pi_{0j}P$ with $E_1\Pi_{0j}w(+\infty) = 0$ can be found from (8), (12) and (15) or (16) in dependence on $j$, and the relations

$$B(0)'(E_2 + E_3)\Pi_{0j}\psi - R(0)\Pi_{0j}u - [\widehat{\Pi}_{0(j-1)}\chi(\tau_0,\varepsilon)]_j = 0, \tag{21}$$

$$(E_1 + E_2)\frac{d\Pi_{0j}\psi}{d\tau_0} = W(0)\Pi_{0j}w - A(0)'(E_2 + E_3)\Pi_{0j}\psi$$

$$+[\widehat{\Pi}_{0(j-1)}\rho(\tau_0,\varepsilon)]_j - E_3\frac{d\Pi_{0(j-1)}\psi}{d\tau_0}, \tag{22}$$

$$(E_1 + E_2)\Pi_{0j}\psi(+\infty) = 0.$$

A solution of the problem $Q_{0j}P$ with $(E_1 + E_2)Q_{0j}w(-\infty) = 0$ can be found from (9), (12) and the relations

$$B(T)'(E_2 + E_3)Q_{0j}\psi - R(T)Q_{0j}u - [\widehat{Q}_{0(j-1)}\chi(\sigma_0,\varepsilon)]_j = 0,$$

$$(E_1 + E_2)\frac{dQ_{0j}\psi}{d\sigma_0} = W(T)Q_{0j}w - A(T)'(E_2 + E_3)Q_{0j}\psi$$

$$+[\widehat{Q}_{0(j-1)}\rho(\sigma_0,\varepsilon)]_j - E_3\frac{dQ_{0(j-1)}\psi}{d\sigma_0},$$

$$E_1Q_{0j}\psi(-\infty) = 0, \; E_2Q_{0j}\psi(0) = -E_2(\overline{\psi}_j(T) + Q_{1(j-1)}\psi(0)). \tag{23}$$

A solution of the problem $\Pi_{1j}P$ with $(E_1 + E_2)\Pi_{1j}w(+\infty) = 0$ can be found from (10), (12), (17) in dependence on $j$, and the relations

$$B(0)'E_3\Pi_{1j}\psi - R(0)\Pi_{1j}u - [\widehat{\Pi}_{1(j-1)}\chi(\tau_1,\varepsilon)]_j = 0, \tag{24}$$

$$\frac{d\Pi_{1j}\psi}{d\tau_1} = W(0)\Pi_{1j}w - A(0)'E_3\Pi_{1j}\psi + [\widehat{\Pi}_{1(j-1)}\rho(\tau_1,\varepsilon)]_j, \tag{25}$$

$$\Pi_{1j}\psi(+\infty) = 0.$$

A solution of the problem $Q_{1j}P$ with $Q_{1j}w(-\infty) = 0$ can be found from (11), (12) in dependence on $j$, and the relations

$$B(T)'E_3Q_{1j}\psi - R(T)Q_{1j}u - [\widehat{Q}_{1(j-1)}\chi(\sigma_1,\varepsilon)]_j = 0,$$

$$\frac{dQ_{1j}\psi}{d\sigma_1} = W(T)Q_{1j}w - A(T)'E_3Q_{1j}\psi + [\widehat{Q}_{1(j-1)}\rho(\sigma_1,\varepsilon)]_j,$$

$$(E_1 + E_2)Q_{1j}\psi(-\infty) = 0, \; E_3 Q_{1j}\psi(0) = -E_3(\overline{\psi}_j(T) + Q_{0j}\psi(0)). \tag{26}$$

In view of the control optimality condition in the Pontryagin maximum principle, a solution of the problem (1)–(3) satisfies (2), (3) and the following relations, including the costate variable $\varphi(t,\varepsilon) = (\zeta(t,\varepsilon)', \eta(t,\varepsilon)', \theta(t,\varepsilon)')'$,

$$B(t)'\varphi - R(t)u - \varepsilon(F_u(\vartheta,t,\varepsilon)' - f_u(\vartheta,t,\varepsilon)'\varphi) = 0, \tag{27}$$

$$\mathcal{E}(\varepsilon)\frac{d\varphi}{dt} = W(t)w - A(t)'\varphi + \varepsilon(F_w(\vartheta,t,\varepsilon)' - f_w(\vartheta,t,\varepsilon)'\varphi), \tag{28}$$

$$\varphi(T,\varepsilon) = 0. \tag{29}$$

An asymptotic solution of problems (2), (3), (27)–(29) can be constructed in the form (4), i.e., in addition, we set

$$\varphi(t,\varepsilon) = \overline{\varphi}(t,\varepsilon) + \sum_{i=0}^{1}(\Pi_i\varphi(\tau_i,\varepsilon) + Q_i\varphi(\sigma_i,\varepsilon)), \tag{30}$$

where all terms have the properties of the corresponding terms in (4).

Substitute asymptotic expansions (4), (30) into (27)–(29) and use presentation (5). Introducing the notation $g(\vartheta,\varphi,t,\varepsilon) = \rho(\vartheta,\varphi,t,\varepsilon)$, $h(\vartheta,\varphi,t,\varepsilon) = \chi(\vartheta,\varphi,t,\varepsilon)$ and equating terms of the same power of $\varepsilon$ separately depending on $t$, $\tau_i$, $\sigma_i$, $i = 0,1$, we obtain the relations

$$B(t)'\overline{\varphi}_j - R(t)\overline{u}_j - [\widehat{\overline{h}}_{j-1}(t,\varepsilon)]_j = 0,$$

$$E_1\frac{d\overline{\varphi}_j}{dt} + E_2\frac{d\overline{\varphi}_{j-1}}{dt} + E_3\frac{d\overline{\varphi}_{j-2}}{dt} = W(t)\overline{w}_j - A(t)'\overline{\varphi}_j + [\widehat{\overline{g}}_{j-1}(t,\varepsilon)]_j,$$

$$B(0)'\Pi_{ij}\varphi - R(0)\Pi_{ij}u - [\widehat{\Pi}_{i(j-1)}h(\tau_i,\varepsilon)]_j = 0,$$

$$\begin{aligned}E_1\frac{d\Pi_{ij}\varphi}{d\tau_i} + E_2\frac{d\Pi_{i(j-1)}\varphi}{d\tau_i} + E_3\frac{d\Pi_{i(j-2)}\varphi}{d\tau_i} &= W(0)\Pi_{i(j-i-1)}w \\ -A(0)'\Pi_{i(j-i-1)}\varphi + [\widehat{\Pi}_{i(j-i-2)}g(\tau_i,\varepsilon)]_{j-i-1},\end{aligned} \tag{31}$$

$$B(T)'Q_{ij}\varphi - R(T)Q_{ij}u - [\widehat{Q}_{i(j-1)}h(\sigma_i,\varepsilon)]_j = 0,$$

$$\begin{aligned}E_1\frac{dQ_{ij}\varphi}{d\sigma_i} + E_2\frac{dQ_{i(j-1)}\varphi}{d\sigma_i} + E_3\frac{dQ_{i(j-2)}\varphi}{d\sigma_i} &= W(T)Q_{i(j-i-1)}w \\ -A(T)'Q_{i(j-i-1)}\varphi + [\widehat{Q}_{i(j-i-2)}g(\sigma_i,\varepsilon)]_{j-i-1},\end{aligned} \tag{32}$$

$$\overline{\varphi}_j(T) + Q_{0j}\varphi(0) + Q_{1j}\varphi(0) = 0. \tag{33}$$

It follows from (31), (32) with $j = 0$ and $i = j = 1$ that

$$E_1\Pi_{00}\varphi(\tau_0) = 0, \; E_1\Pi_{10}\varphi(\tau_1) = E_1\Pi_{11}\varphi(\tau_1) = 0, \; E_1 Q_{00}\varphi(\sigma_0) = 0,$$
$$E_1 Q_{10}\varphi(\sigma_1) = E_1 Q_{11}\varphi(\sigma_1) = 0, \; E_2\Pi_{10}\varphi(\tau_1) = 0, \; E_2 Q_{10}\varphi(\sigma_1) = 0.$$

## 4. Justification of Formalism of Asymptotics Construction

This section deals with the establishment of a relation between the forms of coefficients in the expansion (6) of the minimized functional with respect to powers of $\varepsilon$ and the expressions of the performance indices in optimal control problems formulated in the previous section. The following theorem, which was given in [26] without any rigorous proof, will be further justified.

**Theorem 1.** *The sum* $\overline{J}_j + \Pi_{1(j-1)}J + Q_{1(j-1)}J$ *of the performance indices in problems* $\overline{P}_j$, $\Pi_{1(j-1)}P$, $Q_{1(j-1)}P$ *is obtained by transforming the coefficient* $J_{2j}$ *in expansion (6) and dropping terms, which*

are known after solving problems $\overline{P}_k$, $\Pi_{0k}P$, $Q_{0k}P$, $k = \overline{0, j-1}$, $\Pi_{1k}P$, $Q_{1k}P$, $k = \overline{0, k-2}$. *The sum* $\Pi_{0j}J + Q_{0j}J$ *of the performance indices in problems* $\Pi_{0j}P$, $Q_{0j}P$ *is obtained by transforming the coefficient* $J_{2j+1}$ *in expansion* (6) *and dropping terms, which are known after solving problems* $\overline{P}_k$, $k = \overline{0, j}$, $\Pi_{ik}P$, $Q_{ik}P$, $i = 0, 1$, $k = \overline{0, j-1}$.

**Proof.** Denote the integrand in (1) by means $\mathbb{F}(\vartheta, t, \varepsilon)$. In view of (5), we can present $J_k$ in the form

$$J_k = \int_0^T \overline{\mathbb{F}}_k(t)\, dt + \int_0^{+\infty} \Pi_{0(k-1)}\mathbb{F}(\tau_0)\, d\tau_0$$
$$+ \int_{-\infty}^0 Q_{0(k-1)}\mathbb{F}(\sigma_0)d\sigma_0 + \int_0^{+\infty} \Pi_{1(k-2)}\mathbb{F}(\tau_1)d\tau_1 + \int_{-\infty}^0 Q_{1(k-2)}\mathbb{F}(\sigma_1)\, d\sigma_1. \tag{34}$$

It is clear that the last expression contains the asymptotics terms with numbers more than it is necessary in this theorem, for instance, $\overline{\mathbb{F}}_{2n}(t) = [\mathbb{F}(\widetilde{\overline{\vartheta}}_{2n}(t, \varepsilon), t, \varepsilon)]_{2n}$. In order to prove the theorem, we will use control optimality conditions for formulated previously control problems.

It is evident that the coefficient $J_0$ in (6) is the performance index in problem $\overline{P}_0$.

We will analyze the coefficient $J_1$. In view of (34) with $k = 1$, we have

$$J_1 = \int_0^T \overline{\mathbb{F}}_1(t)\, dt + \int_0^{+\infty} \Pi_{00}\mathbb{F}(\tau_0)\, d\tau_0 + \int_{-\infty}^0 Q_{00}\mathbb{F}(\sigma_0)\, d\sigma_0$$
$$= \int_0^T (\overline{w}_1(t)'W(t)\overline{w}_0(t) + \overline{u}_1(t)'R(t)\overline{u}_0(t) + [\widehat{\overline{\mathbb{F}}}_0(t, \varepsilon)]_1)\, dt$$
$$+ \int_0^{+\infty} \frac{1}{2}(\Pi_{00}w(\tau_0)'W(0)\Pi_{00}w(\tau_0) + \Pi_{00}u(\tau_0)'R(0)\Pi_{00}u(\tau_0))$$
$$+ \Pi_{00}w(\tau_0)'W(0)\overline{w}_0(0) + \Pi_{00}u(\tau_0)'R(0)\overline{u}_0(0))\, d\tau_0$$
$$+ \int_{-\infty}^0 \frac{1}{2}(Q_{00}w(\sigma_0)'W(T)Q_{00}w(\sigma_0) + Q_{00}u(\sigma_0)'R(T)Q_{00}u(\sigma_0))$$
$$+ Q_{00}w(\sigma_0)'W(T)\overline{w}_0(T) + Q_{00}u(\sigma_0)'R(T)\overline{u}_0(T))\, d\sigma_0.$$

Transforming the following expression from $J_1$ with the help of control optimality conditions for the problem $\overline{P}_0$ (see (18)–(20) with $j = 0$), the integration by parts, and also (12), (7) with $j = 1$, (8), (9) with $j = 0$ and $j = 1$, and (15), we have

$$\int_0^T (\overline{w}_1(t)'W(t)\overline{w}_0(t) + \overline{u}_1(t)'R(t)\overline{u}_0(t))\, dt + \int_0^{+\infty} (\Pi_{00}w(\tau_0)'W(0)\overline{w}_0(0)$$
$$+ \Pi_{00}u(\tau_0)'R(0)\overline{u}_0(0))\, d\tau_0 + \int_{-\infty}^0 (Q_{00}w(\sigma_0)'W(T)\overline{w}_0(T) + Q_{00}u(\sigma_0)'R(T)\overline{u}_0(T))\, d\sigma_0$$
$$= \int_0^T (\overline{w}_1(t)'(E_1 \frac{d\overline{\psi}_0(t)}{dt} + A(t)'\overline{\psi}_0(t)) + \overline{u}_1(t)'B(t)'\overline{\psi}_0(t))\, dt$$
$$+ \int_0^{+\infty} (\Pi_{00}w(\tau_0)'(E_1 \frac{d\overline{\psi}_0}{dt}(0) + A(0)'\overline{\psi}_0(0)) + \Pi_{00}u(\tau_0)'B(0)'\overline{\psi}_0(0))\, d\tau_0$$
$$+ \int_{-\infty}^0 (Q_{00}w(\sigma_0)'(E_1 \frac{d\overline{\psi}_0}{dt}(T) + A(T)'\overline{\psi}_0(T)) + Q_{00}u(\sigma_0)'B(T)'\overline{\psi}_0(T))\, d\sigma_0$$
$$= \overline{w}_1(t)'E_1\overline{\psi}_0(t)|_0^T + \int_0^T \overline{\psi}_0(t)'(-E_1 \frac{d\overline{w}_1}{dt}(t) + A(t)\overline{w}_1(t) + B(t)\overline{u}_1(t))\, dt$$
$$+ \int_0^{+\infty} \overline{\psi}_0(0)'(A(0)\Pi_{00}w(\tau_0) + B(0)\Pi_{00}u(\tau_0))\, d\tau_0$$
$$+ \int_{-\infty}^0 \overline{\psi}_0(T)'(A(T)Q_{00}w(\sigma_0) + B(T)Q_{00}u(\sigma_0))\, d\sigma_0$$
$$= \Pi_{01}w(0)'E_1\overline{\psi}_0(0) + \int_0^T \overline{\psi}_0(t)'(E_2 \frac{d\overline{w}_0}{dt}(t) - [\widehat{\overline{\phi}}_0(t, \varepsilon)]_1)\, dt$$

$$+ \int_0^{+\infty} \overline{\psi}_0(0)'(E_1 \frac{d\Pi_{01}w(\tau_0)}{d\tau_0} + E_2 \frac{d\Pi_{00}w(\tau_0)}{d\tau_0}) \, d\tau_0$$

$$+ \int_{-\infty}^0 \overline{\psi}_0(T)'(E_1 \frac{dQ_{01}w(\sigma_0)}{d\sigma_0} + E_2 \frac{dQ_{00}w(\sigma_0)}{d\sigma_0}) \, d\sigma_0$$

$$= \Pi_{01}w(0)'E_1\overline{\psi}_0(0) + \int_0^T \overline{\psi}_0(t)'(E_2 \frac{d\overline{w}_0}{dt}(t) - [\widehat{\overline{\phi}}_0(t,\varepsilon)]_1) \, dt$$

$$-\overline{\psi}_0(0)'(E_1\Pi_{01}w(0) + E_2\Pi_{00}w(0)) + \overline{\psi}_0(T)'E_2Q_{00}w(0)$$

$$= \overline{\psi}_0(T)'E_2Q_{00}w(0) + \int_0^T \overline{\psi}_0(t)'(E_2 \frac{d\overline{w}_0}{dt}(t) - [\widehat{\overline{\phi}}_0(t,\varepsilon)]_1) \, dt - \overline{\psi}_0(0)'E_2(w^0 - \overline{w}_0(0)).$$

Taking into account this relation and the previous expression for $J_1$, and also dropping terms, which are known after solving the problem $\overline{P}_0$, we see that the transformed expression for $J_1$ is the sum $\Pi_{00}J + Q_{00}J$.

Assuming that the problems $\overline{P}_0$, $\Pi_{00}P$, $Q_{00}P$ have been solved, we transform by similar way the coefficient $J_2$ in (6). According to (34), $J_2$ has the form:

$$\int_0^T \overline{\mathbb{F}}_2(t) \, dt + \int_0^{+\infty} \Pi_{01}\mathbb{F}(\tau_0) \, d\tau_0 + \int_{-\infty}^0 Q_{01}\mathbb{F}(\sigma_0) \, d\sigma_0$$

$$+ \int_0^{+\infty} \Pi_{10}\mathbb{F}(\tau_1) \, d\tau_1 + \int_{-\infty}^0 Q_{10}\mathbb{F}(\sigma_1) \, d\sigma_1.$$

Write down the unknown terms in $\overline{\mathbb{F}}_2(t)$

$$\overline{w}_2(t)'W(t)\overline{w}_0(t) + \overline{u}_2(t)'R(t)\overline{u}_0(t) + \overline{w}_1(t)'(1/2W(t)\overline{w}_1(t) + \overline{F}_{w_0}(t)')$$

$$+\overline{u}_1(t)'(1/2R(t)\overline{u}_1(t) + \overline{F}_{u_0}(t)').$$

Transforming $\int_0^T (\overline{w}_2'W\overline{w}_0 + \overline{u}_2'R\overline{u}_0) \, dt$ with the help of optimality conditions (18), (19) at $j = 0$, integrating by parts, (7) at $j = 2$, (20) at $j = 0$, and dropping known terms, we obtain $-\overline{\psi}_0(0)'(E_1\overline{w}_2(0) + E_2\overline{w}_1(0)) + \overline{\psi}_0(T)'E_2\overline{w}_1(T) - \int_0^T (\overline{w}_1'E_2d\overline{\psi}_0/dt + \overline{\psi}_0'(\overline{f}_{w_0}\overline{w}_1 + \overline{f}_{u_0}\overline{u}_1)) \, dt$.

The unknown expression in $\Pi_{01}\mathbb{F}(\tau_0)$ is

$$\Pi_{01}w(\tau_0)'W(0)(\overline{w}_0(0) + \Pi_{00}w(\tau_0)) + \Pi_{00}w(\tau_0)'W(0)\overline{w}_1(0)$$

$$+\Pi_{01}u(\tau_0)'R(0)(\overline{u}_0(0) + \Pi_{01}u(\tau_0)) + \Pi_{00}u(\tau_0)'R(0)\overline{u}_1(0).$$

The integral of this expression will be transformed using control optimality conditions for problems $\overline{P}_0$ and $\Pi_{00}P$, Equations (7) at $j = 1$, (8) at $j = 1, 2$, the formula of integration by parts and Remark 1. Dropping known terms, we have $-\Pi_{00}\psi(0)'((E_1 + E_2)\overline{w}_1(0) + E_2\Pi_{01}w(0)) - \overline{\psi}_0(0)'(E_2\Pi_{01}w(0) + E_1\Pi_{02}w(0))$.

Similarly, we transform the third integral in $J_2$, depending on an unknown expression

$$\int_{-\infty}^0 (Q_{01}w(\sigma_0)'W(T)(\overline{w}_0(T) + Q_{00}w(\sigma_0)) + Q_{00}w(\sigma_0)'W(T)\overline{w}_1(T)$$

$$+Q_{01}u(\sigma_0)'R(T)(\overline{u}_0(T) + Q_{00}u(\sigma_0)) + Q_{00}u(\sigma_0)'R_0(T)\overline{u}_1(T)) \, d\sigma_0$$

$$= Q_{00}\psi(0)'((E_1 + E_2)\overline{w}_1(T) + E_2Q_{01}w(0)) + \overline{\psi}_0(T)'E_2Q_{01}w(0).$$

The unknown expression in $\Pi_{10}\mathbb{F}(\tau_1)$ is

$$\Pi_{10}w(\tau_1)'W(0)(\overline{w}_0(0) + \Pi_{00}w(0)) + \Pi_{10}u(\tau_1)'R(0)(\overline{u}_0(0) + \Pi_{00}u(0))$$

$$+1/2(\Pi_{10}w(\tau_1)'W(0)\Pi_{10}w(\tau_1) + \Pi_{10}u(\tau_1)'R(0)\Pi_{10}u(\tau_1)).$$

Transform the integral

$$\int_0^{+\infty} (\Pi_{10}w(\tau_1)'W(0)(\overline{w}_0(0) + \Pi_{00}w(0)) + \Pi_{10}u(\tau_1)'R(0)(\overline{u}_0(0) + \Pi_{00}u(0)))\, d\tau_1$$

with the help of optimality conditions for problems $\overline{P}_0$, $\Pi_{00}P$, (10) at $j = 0, 1, 2$, (12) and integration by parts. Dropping known terms, we have $-\overline{\psi}_0(0)'(E_1\Pi_{12}w(0) + E_2\Pi_{11}w(0) + E_3\Pi_{10}w(0)) - \Pi_{00}\psi(0)'(E_2\Pi_{11}w(0) + E_3\Pi_{10}w(0))$.

Transforming in a similar way the fifth integral in $J_2$, depending on unknown terms, we obtain

$$\int_{-\infty}^0 (Q_{10}w(\sigma_1)'W(T)(\overline{w}_0(T) + Q_{00}w(0)) + Q_{10}u(\sigma_1)'R(T)(\overline{u}_0(T) + Q_{00}u(0))$$
$$+ 1/2(Q_{10}w(\sigma_1)'W(T)Q_{10}w(\sigma_1) + Q_{10}u(\sigma_1)'R(T)Q_{10}u(\sigma_1)))\, d\sigma_1$$
$$= \overline{\psi}_0(T)'(E_2Q_{11}w(0) + E_3Q_{10}w(0)) + Q_{00}\psi(0)'(E_2Q_{11}w(0) + E_3Q_{10}w(0))$$
$$+ 1/2 \int_{-\infty}^0 (Q_{10}w(\sigma_1)'W(T)Q_{10}w(\sigma_1) + Q_{10}u(\sigma_1)'R(T)Q_{10}u(\sigma_1))\, d\sigma_1.$$

Substituting the transformed relations into $J_2$, taking into account the second equality in (13), (14) at $j = 2$, (16) at $j = 1$, (17) and (23) at $j = 0$, and also Remark 1, and finally dropping known terms, we obtain the theorem statement for the coefficient $J_2$.

Introduce the notation

$$\vartheta(t,\varepsilon) - \widetilde{\vartheta}_{j-1}(t,\varepsilon) = \Delta_j\overline{\vartheta}(t,\varepsilon) + \sum_{i=0}^1 (\Delta_j\Pi_i\vartheta(\tau_i,\varepsilon) + \Delta_jQ_i\vartheta(\sigma_i,\varepsilon)), \tag{35}$$

where $\Delta_j\overline{\vartheta}(t,\varepsilon) = \overline{\vartheta}(t,\varepsilon) - \widetilde{\overline{\vartheta}}_{j-1}(t,\varepsilon) = \varepsilon^j\overline{\vartheta}_j(t) + \alpha(\varepsilon^{j+1})$, $\Delta_j\Pi_i\vartheta(\tau_i,\varepsilon) = \Pi_i\vartheta(\tau_i,\varepsilon) - \widetilde{\Pi}_{i(j-1)}\vartheta(\tau_i,\varepsilon) = \varepsilon^j\Pi_{ij}\vartheta(\tau_i) + \alpha(\varepsilon^{j+1})$, $\Delta_jQ_i\vartheta(\sigma_i,\varepsilon) = Q_i\vartheta(\tau_i,\varepsilon) - \widetilde{Q}_{i(j-1)}\vartheta(\sigma_i,\varepsilon) = \varepsilon^jQ_{ij}\vartheta(\sigma_i) + \alpha(\varepsilon^{j+1})$, $i = 0, 1$, $\alpha(\varepsilon^{j+1})$ is a sum of the expansion terms of order $\varepsilon^{j+1}$ and higher.

Assuming that the problems $\overline{P}_j$, $\Pi_{0j}P$, $Q_{0j}P$ and $\Pi_{1(j-1)}P$, $Q_{1(j-1)}P$, $j = \overline{0, n-1}$ have been solved, we will transform each term in the coefficient $J_{2n}$, having the presentation (34) with $k = 2n$.

Using the notation (35), we can see that the unknown terms in $\overline{\mathbb{F}}_{2n}(t)$ are the following:

$$\overline{w}_n(t)'(1/2W(t)\overline{w}_n(t) + [\widehat{\overline{F}}_{w(n-1)}(t,\varepsilon)']_{n-1})$$
$$+ \overline{u}_n(t)'(1/2R(t)\overline{u}_n(t) + [\widehat{\overline{F}}_{u(n-1)}(t,\varepsilon)']_{n-1})$$
$$+ [\Delta_{n+1}\overline{w}(t,\varepsilon)'(W(t)\widetilde{\overline{w}}_{n-1}(t,\varepsilon) + \{\varepsilon\widehat{\overline{F}}_{w(n-1)}(t,\varepsilon)'\}_{n-1})]_{2n}$$
$$+ [\Delta_{n+1}\overline{u}(t,\varepsilon)'(R(t)\widetilde{\overline{u}}_{n-1}(t,\varepsilon) + \{\varepsilon\widehat{\overline{F}}_{u(n-1)}(t,\varepsilon)'\}_{n-1})]_{2n}.$$

Multiplying the Equations (18), (19) by $\varepsilon^j$, $j = \overline{0, k}$, and summing up the obtained equations, we obtain the following relations

$$\{R(t)\widetilde{\overline{u}}_k(t,\varepsilon) + \varepsilon\widehat{\overline{F}}_{u(k-1)}(t,\varepsilon)'\}_k = \{B(t)'\widetilde{\overline{\psi}}_k(t,\varepsilon) + \varepsilon\widehat{\widetilde{f}}_{u(k-1)}(t,\varepsilon)'\widetilde{\overline{\psi}}_{k-1}(t,\varepsilon)\}_k,$$

$$\{W(t)\widetilde{\overline{w}}_k(t,\varepsilon) + \varepsilon\widehat{\overline{F}}_{w(k-1)}(t,\varepsilon)'\}_k = \{\mathcal{E}(\varepsilon)\frac{d\widetilde{\overline{\psi}}_k(t,\varepsilon)}{dt}\}_k \tag{36}$$

$$+ \{A(t)'\widetilde{\overline{\psi}}_k(t,\varepsilon) + \varepsilon\widehat{\widetilde{f}}_{w(k-1)}(t,\varepsilon)'\widetilde{\overline{\psi}}_{k-1}(t,\varepsilon)\}_k.$$

Substituting $\vartheta(t, \varepsilon)$ from (35) with $j = n + 1$ into (2) and equating terms depending on $t$, we obtain the equation

$$\mathcal{E}(\varepsilon)\left(\frac{d\widetilde{\overline{w}}_n(t, \varepsilon)}{dt} + \frac{d\Delta_{n+1}\overline{w}(t, \varepsilon)}{dt}\right) = A(t)(\widetilde{\overline{w}}_n(t, \varepsilon) + \Delta_{n+1}\overline{w}(t, \varepsilon))$$

$$+ B(t)(\widetilde{\overline{u}}_n(t, \varepsilon) + \Delta_{n+1}\overline{u}(t, \varepsilon)) + \varepsilon f(\widetilde{\overline{\vartheta}}_n(t, \varepsilon) + \Delta_{n+1}\overline{\vartheta}(t, \varepsilon), t, \varepsilon). \tag{37}$$

We will use the next easily proved formula from [29], which is valid for any sufficiently smooth vector functions $a(t, \varepsilon)$, $b(t, \varepsilon)$ and a matrix $\mathbb{D}(t, \varepsilon)$ of the corresponding size,

$$_k = [\{b(t, \varepsilon)\}'_l \mathbb{D}(t, \varepsilon) a(t, \varepsilon)]_k$$

$$- [\{b(t, \varepsilon)\}'_l \mathbb{D}(t, \varepsilon)\{a(t, \varepsilon)\}_{k-l-1}]_k, \; k, l \in IN, \; k \geqslant l. \tag{38}$$

Using (36) with $k = n - 1$, (37), (38) with $l = n - 1$, $k = 2n$, we can rewrite

$$\int_0^T ([\Delta_{n+1}\overline{w}(t, \varepsilon)'(W(t)\widetilde{\overline{w}}_{n-1}(t, \varepsilon) + \{\varepsilon\widehat{\overline{F}}_{w(n-1)}(t, \varepsilon)'\}_{n-1})]_{2n}$$

$$+ [\Delta_{n+1}\overline{u}(t, \varepsilon)'(R(t)\widetilde{\overline{u}}_{n-1}(t, \varepsilon) + \{\varepsilon\widehat{\overline{F}}_{u(n-1)}(t, \varepsilon)'\}_{n-1})]_{2n})\, dt$$

in the following way

$$\int_0^T ([\Delta_{n+1}\overline{w}(t, \varepsilon)'\{\mathcal{E}(\varepsilon)\frac{d\widetilde{\overline{\psi}}_{n-1}(t, \varepsilon)}{dt}\}_{n-1}]_{2n} + [\widetilde{\overline{\psi}}_{n-1}(t, \varepsilon)'(A(t)\Delta_{n+1}\overline{w}(t, \varepsilon)$$

$$+ B(t)\Delta_{n+1}\overline{u}(t, \varepsilon)]_{2n} + [\Delta_{n+1}\overline{w}(t, \varepsilon)'\{\varepsilon\widehat{\overline{f}}_{w(n-1)}(t, \varepsilon)'\widetilde{\overline{\psi}}_{n-1}(t, \varepsilon)\}_{n-1}]_{2n}$$

$$+ [\Delta_{n+1}\overline{u}(t, \varepsilon)'\{\varepsilon\widehat{\overline{f}}_{u(n-1)}(t, \varepsilon)'\widetilde{\overline{\psi}}_{n-1}(t, \varepsilon)\}_{n-1}\rangle]_{2n})\, dt$$

$$= \int_0^T ([\Delta_{n+1}\overline{w}(t, \varepsilon)'\{\mathcal{E}(\varepsilon)\frac{d\widetilde{\overline{\psi}}_{n-1}(t, \varepsilon)}{dt}\}_{n-1}]_{2n} + [\widetilde{\overline{\psi}}_{n-1}(t, \varepsilon)'(\mathcal{E}(\varepsilon)(\frac{d\Delta_{n+1}\overline{w}(t, \varepsilon)}{dt}$$

$$+ \frac{d\widetilde{\overline{w}}_n(t, \varepsilon)}{dt}) - A(t)\widetilde{\overline{w}}_n(t, \varepsilon) - B(t)\widetilde{\overline{u}}_n(t, \varepsilon) - \varepsilon f(\widetilde{\overline{\vartheta}}_n(t, \varepsilon) + \Delta_{n+1}\overline{\vartheta}(t, \varepsilon), t, \varepsilon))]_{2n}$$

$$+ [\widetilde{\overline{\psi}}_{n-1}(t, \varepsilon)'(\{\varepsilon\widehat{\overline{f}}_{w(n-1)}(t, \varepsilon)\}_{n-1}\Delta_{n+1}\overline{w}(t, \varepsilon) + \{\varepsilon\widehat{\overline{f}}_{u(n-1)}(t, \varepsilon)\}_{n-1}\Delta_{n+1}\overline{u}(t, \varepsilon))]_{2n})\, dt.$$

Integrating by parts in the first term of the last expression, taking into account the equality $\Delta_{n+1}\overline{\vartheta}(t, \varepsilon) = \Delta_n\overline{\vartheta}(t, \varepsilon) - \varepsilon^n\overline{\vartheta}_n(t)$, decomposing $f(\widetilde{\overline{\vartheta}}_n(t, \varepsilon) + \Delta_{n+1}\overline{\vartheta}(t, \varepsilon), t, \varepsilon)$ in the neighborhood of $\widetilde{\overline{\vartheta}}_{n-1}(t, \varepsilon)$, and omitting known terms, we obtain

$$[\Delta_{n+1}\overline{w}(t, \varepsilon)'\mathcal{E}(\varepsilon)\widetilde{\overline{\psi}}_{n-1}(t, \varepsilon)]_{2n}\big|_0^T + (\overline{\psi}_{n-1}(t)'E_2\overline{w}_n(t) + \overline{\psi}_{n-2}(t)'E_3\overline{w}_n(t))\big|_0^T$$

$$+ \int_0^T (\overline{w}'_n(-E_2\frac{d\overline{\psi}_{n-1}}{dt} - E_3\frac{d\overline{\psi}_{n-2}}{dt} - [\{\widetilde{\overline{\psi}}_{n-1}(t, \varepsilon)'\varepsilon\widehat{\overline{f}}_{\vartheta(n-1)}(t, \varepsilon)\}_n(\varepsilon^n\overline{\vartheta}_n + \Delta_{n+1}\vartheta(t, \varepsilon))]_{2n}$$

$$+ [\widetilde{\overline{\psi}}_{n-1}(t, \varepsilon)'(\{\varepsilon\widehat{\overline{f}}_{w(n-1)}(t, \varepsilon)\}_{n-1}\Delta_{n+1}\overline{w}(t, \varepsilon) + \{\varepsilon\widehat{\overline{f}}_{u(n-1)}(t, \varepsilon)\}_{n-1}\Delta_{n+1}\overline{u}(t, \varepsilon))]_{2n})\, dt$$

$$= [\Delta_n\overline{w}(t, \varepsilon)'(\frac{1}{\varepsilon}E_1 + E_2 + \varepsilon E_3)\widetilde{\overline{\psi}}_{n-1}(t, \varepsilon)]_{2n-1}\big|_0^T$$

$$- \int_0^T (\overline{w}_n(t)'(E_2\frac{d\overline{\psi}_{n-1}}{dt} + E_3\frac{d\overline{\psi}_{n-2}}{dt} + [\varepsilon\widehat{\overline{f}}_{w(n-1)}(t, \varepsilon)'\widetilde{\overline{\psi}}_{n-1}(t, \varepsilon)]_n)$$

$$+ [\overline{u}_n(t)'([\varepsilon\widehat{\overline{f}}_{u(n-1)}(t, \varepsilon)'\widetilde{\overline{\psi}}_{n-1}(t, \varepsilon)]_n)\, dt.$$

Taking into account the last relation, omitting known terms, we obtain the following expression for the first term of $J_{2n}$:

$$\int_0^T \overline{\mathbb{F}}_{2n}(t)\, dt = [\Delta_n\overline{w}(t, \varepsilon)'(\frac{1}{\varepsilon}E_1 + E_2 + \varepsilon E_3)\widetilde{\overline{\psi}}_{n-1}(t, \varepsilon)]_{2n-1}\big|_0^T + \overline{J}_n$$

$$-\overline{w}_n(T)'E_1(Q_{0(n-1)}\psi(0) + Q_{1(n-2)}\psi(0)).$$

The next step is the transformation of the unknown parts of $\Pi_{0(2n-1)}\mathbb{F}(\tau_0)$, which, after substituting (35) and some transformations, is given below

$$[\Delta_n\Pi_0 w(\tau_0,\varepsilon)'(\{W(\varepsilon\tau_0)\widetilde{\overline{w}}_{n-1}(\varepsilon\tau_0,\varepsilon)\}_{n-1} + \{\varepsilon\widehat{\overline{F}}_{w(n-1)}(\varepsilon\tau_0,\varepsilon)'\}_{n-1})]_{2n-1}$$
$$+[\Delta_n\Pi_0 u(\tau_0,\varepsilon)'(\{R(\varepsilon\tau_0)\widetilde{\overline{u}}_{n-1}(\varepsilon\tau_0,\varepsilon)\}_{n-1} + \{\varepsilon\widehat{\overline{F}}_{u(n-1)}(\varepsilon\tau_0,\varepsilon)'\}_{n-1})]_{2n-1}$$
$$+[(\Delta_n\overline{w}(\varepsilon\tau_0,\varepsilon) + \Delta_n\Pi_0 w(\tau_0,\varepsilon))'(\{W(\varepsilon\tau_0)\widetilde{\Pi}_{0(n-1)}w(\tau_0,\varepsilon)\}_{n-1}$$
$$+\{\varepsilon\widehat{\Pi}_{0(n-1)}F_w(\tau_0,\varepsilon)'\}_{n-1})]_{2n-1} + [(\Delta_n\overline{u}(\varepsilon\tau_0,\varepsilon)$$
$$+\Delta_n\Pi_0 u(\tau_0,\varepsilon))'(\{R(\varepsilon\tau_0)\widetilde{\Pi}_{0(n-1)}u(\tau_0,\varepsilon)\}_{n-1} + \{\varepsilon\widehat{\Pi}_{0(n-1)}F_u(\tau_0,\varepsilon)'\}_{n-1})]_{2n-1}.$$

Substituting $\vartheta(t,\varepsilon)$ from (35) into (2) and considering terms depending on $\tau_0$, we obtain the equation

$$(\frac{1}{\varepsilon}E_1 + E_2 + \varepsilon E_3)(\frac{d\widetilde{\Pi}_{0(n-1)}w(\tau_0,\varepsilon)}{d\tau_0} + \frac{d\Delta_n\Pi_0 w(\tau_0,\varepsilon)}{d\tau_0})$$
$$= A(\varepsilon\tau_0)(\widetilde{\Pi}_{0(n-1)}w(\tau_0,\varepsilon) + \Delta_n\Pi_0 w(\tau_0,\varepsilon)) + B(\varepsilon\tau_0)(\widetilde{\Pi}_{0(n-1)}u(\tau_0,\varepsilon) \qquad (39)$$
$$+\Delta_n\Pi_0 u(\tau_0,\varepsilon)) + \varepsilon(f(\widetilde{\overline{\vartheta}}_{n-1}(\varepsilon\tau_0,\varepsilon) + \widetilde{\Pi}_{0(n-1)}\vartheta(\tau_0,\varepsilon)$$
$$+\Delta_n\overline{\vartheta}(\varepsilon\tau_0,\varepsilon) + \Delta_n\Pi_0\vartheta(\tau_0,\varepsilon),\varepsilon\tau_0,\varepsilon) - f(\widetilde{\overline{\vartheta}}_{n-1}(\varepsilon\tau_0,\varepsilon) + \Delta_n\overline{\vartheta}(\varepsilon\tau_0,\varepsilon),\varepsilon\tau_0,\varepsilon)).$$

Using (36) with $k = n - 1$, (39) and (38), we transform the following expression:

$$\int_0^{+\infty}([\Delta_n\Pi_0 w(\tau_0,\varepsilon)'(\{W(\varepsilon\tau_0)\widetilde{\overline{w}}_{n-1}(\varepsilon\tau_0,\varepsilon)\}_{n-1} + \{\varepsilon\widehat{\overline{F}}_{w(n-1)}(\varepsilon\tau_0,\varepsilon)'\}_{n-1})]_{2n-1}$$
$$+[\Delta_n\Pi_0 u(\tau_0,\varepsilon)'(\{R(\varepsilon\tau_0)\widetilde{\overline{u}}_{n-1}(\varepsilon\tau_0,\varepsilon)\}_{n-1} + \{\varepsilon\widehat{\overline{F}}_{u(n-1)}(\varepsilon\tau_0,\varepsilon)'\}_{n-1})]_{2n-1})\, d\tau_0.$$

Omitting known terms, we have

$$\int_0^{+\infty}([\Delta_n\Pi_0 w(\tau_0,\varepsilon)'(\{\mathcal{E}(\varepsilon)\frac{d\widetilde{\overline{\psi}}_{n-1}}{dt}(\varepsilon\tau_0,\varepsilon)\}_{n-1} + \{A(\varepsilon\tau_0)'\widetilde{\overline{\psi}}_{n-1}(\varepsilon\tau_0,\varepsilon)\}_{n-1}$$
$$+\{\varepsilon\widehat{\overline{f}}_{w(n-1)}(\varepsilon\tau_0,\varepsilon)'\widetilde{\overline{\psi}}_{n-1}(\varepsilon\tau_0,\varepsilon)\}_{n-1})]_{2n-1} + [\Delta_n\Pi_0 u(\tau_0,\varepsilon)'(\{B(\varepsilon\tau_0)'\widetilde{\overline{\psi}}_{n-1}(\varepsilon\tau_0,\varepsilon)\}_{n-1}$$
$$+\{\varepsilon\widehat{\overline{f}}_{u(n-1)}(\varepsilon\tau_0,\varepsilon)'\widetilde{\overline{\psi}}_{n-1}(\varepsilon\tau_0,\varepsilon)\}_{n-1})]_{2n-1})\, d\tau_0 = \int_0^{+\infty}(\Pi_{0n}w'E_1\frac{d\widetilde{\overline{\psi}}_{n-1}}{dt}(0)$$
$$+[\Delta_n\Pi_0 w(\tau_0,\varepsilon)'\{(\frac{1}{\varepsilon}E_1 + E_2 + \varepsilon E_3)\frac{d\widetilde{\overline{\psi}}_{n-2}}{dt}(\varepsilon\tau_0,\varepsilon)\}_{n-2}]_{2n-2}$$
$$+[\widetilde{\overline{\psi}}_{n-1}(\varepsilon\tau_0,\varepsilon)'(\frac{1}{\varepsilon}E_1 + E_2 + \varepsilon E_3)\frac{d\Delta_n\Pi_0 w}{d\tau_0}(\tau_0,\varepsilon)]_{2n-1}$$
$$-[\widetilde{\overline{\psi}}_{n-1}(\varepsilon\tau_0,\varepsilon)'(\varepsilon\widehat{\Pi}_{0(n-1)}f_\vartheta(\tau_0,\varepsilon)\Delta_n\overline{\vartheta}(\varepsilon\tau_0,\varepsilon) + \varepsilon f_\vartheta(\widetilde{\overline{\vartheta}}_{n-1}(\varepsilon\tau_0,\varepsilon)$$
$$+\widetilde{\Pi}_{0(n-1)}\vartheta(\tau_0,\varepsilon),\varepsilon\tau_0,\varepsilon)\Delta_n\Pi_0\vartheta(\tau_0,\varepsilon))]_{2n-1} + [\widetilde{\overline{\psi}}_{n-1}(\varepsilon\tau_0,\varepsilon)'(\varepsilon\widehat{\overline{f}}_{w(n-1)}(\varepsilon\tau_0,\varepsilon)\Delta_n\Pi_0 w(\tau_0,\varepsilon)$$
$$+\varepsilon\widehat{\overline{f}}_{u(n-1)}(\varepsilon\tau_0,\varepsilon)\Delta_n\Pi_0 u(\tau_0,\varepsilon))]_{2n-1})\, d\tau_0.$$

From here, applying the formula of integrating by parts and Remark 1, omitting known terms, we obtain

$$-[\Delta_n\Pi_0 w(0,\varepsilon)'(\frac{1}{\varepsilon}E_1 + E_2 + \varepsilon E_3)\widetilde{\overline{\psi}}_{n-1}(0,\varepsilon)]_{2n-1}$$
$$-\int_0^{+\infty}([\widetilde{\overline{\psi}}_{n-1}(\varepsilon\tau_0,\varepsilon)'(\varepsilon\widehat{\Pi}_{0(n-1)}f_\vartheta(\Delta_n\overline{\vartheta}(\varepsilon\tau_0,\varepsilon) + \Delta_n\Pi_0\vartheta(\tau_0,\varepsilon))]_{2n-1})\, d\tau_0.$$

Multiplying the Equations (21), (22) by $\varepsilon^j$, $j = \overline{0,k}$, and summing up the obtained equations, we obtain the equalities

$$
\begin{aligned}
&\{R(\varepsilon\tau_0)\widetilde{\Pi}_{0k}u\}_k + \{\varepsilon\widehat{\Pi}_{0(k-1)}F_u(\tau_0,\varepsilon)'\}_k \\
&= \{B(\varepsilon\tau_0)'(\varepsilon E_1 + E_2 + E_3)\widetilde{\Pi}_{0k}\psi\}_k + \{\varepsilon\widehat{\Pi}_{0(k-1)}f_u(\tau_0,\varepsilon)'\overline{\widetilde{\psi}}_{k-1}(\varepsilon\tau_0,\varepsilon)\}_k \\
&\quad + \{\varepsilon f_u(\overline{\widetilde{\vartheta}}_{k-1}(\varepsilon\tau_0,\varepsilon) + \widetilde{\Pi}_{0(k-1)}\vartheta(\tau_0,\varepsilon),\varepsilon\tau_0,\varepsilon)'(\varepsilon E_1 + E_2 + E_3)\widetilde{\Pi}_{0(k-1)}\psi\}_k, \\
&\{W(\varepsilon\tau_0)\widetilde{\Pi}_{0k}w\}_k + \{\varepsilon\widehat{\Pi}_{0(k-1)}F_w(\tau_0,\varepsilon)'\}_k = \{(E_1 + E_2 + \varepsilon E_3)\frac{d\widetilde{\Pi}_{0k}\psi}{d\tau_0}\}_k \\
&\quad + \{A(\varepsilon\tau_0)'(\varepsilon E_1 + E_2 + E_3)\widetilde{\Pi}_{0k}\psi\}_k + \{\varepsilon\widehat{\Pi}_{0(k-1)}f_w(\tau_0,\varepsilon)'\overline{\widetilde{\psi}}_{k-1}(\varepsilon\tau_0,\varepsilon)\}_k \\
&\quad + \{\varepsilon f_w(\overline{\widetilde{\vartheta}}_{k-1}(\varepsilon\tau_0,\varepsilon) + \widetilde{\Pi}_{0(k-1)}\vartheta(\tau_0,\varepsilon),\varepsilon\tau_0,\varepsilon)'(\varepsilon E_1 + E_2 + E_3)\widetilde{\Pi}_{0(k-1)}\psi\}_k.
\end{aligned} \tag{40}
$$

We transform

$$
\begin{aligned}
\int_0^{+\infty} &([(\Delta_n\overline{w}(\varepsilon\tau_0,\varepsilon) + \Delta_n\Pi_0 w(\tau_0,\varepsilon))'(\{W(\varepsilon\tau_0)\widetilde{\Pi}_{0(n-1)}w(\tau_0,\varepsilon)\}_{n-1} \\
&\quad + \{\varepsilon\widehat{\Pi}_{0(n-1)}F_w(\tau_0,\varepsilon)'\}_{n-1})]_{2n-1} + [(\Delta_n\overline{u}(\varepsilon\tau_0,\varepsilon) \\
&+ \Delta_n\Pi_0 u(\tau_0,\varepsilon))'(\{R(\varepsilon\tau_0)\widetilde{\Pi}_{0(n-1)}u(\tau_0,\varepsilon)\}_{n-1} + \{\varepsilon\widehat{\Pi}_{0(n-1)}F_u(\tau_0,\varepsilon)'\}_{n-1})]_{2n-1})\,d\tau_0.
\end{aligned}
$$

Using (40) and (38), as a result, we obtain

$$
\begin{aligned}
\int_0^{+\infty} &([(\Delta_n\overline{w}(\varepsilon\tau_0,\varepsilon) + \Delta_n\Pi_0 w(\tau_0,\varepsilon))'\{(E_1 + E_2 + \varepsilon E_3)\frac{d\widetilde{\Pi}_{0(n-1)}\psi}{d\tau_0}\}_{n-1}]_{2n-1} \\
&+ [\widetilde{\Pi}_{0(n-1)}\psi'(\varepsilon E_1 + E_2 + E_3)(A(\varepsilon\tau_0)\Delta_n\overline{w}(\varepsilon\tau_0,\varepsilon) + B(\varepsilon\tau_0)\Delta_n\overline{u}(\varepsilon\tau_0,\varepsilon))]_{2n-1} \\
&+ [\widetilde{\Pi}_{0(n-1)}\psi'(\varepsilon E_1 + E_2 + E_3)(A(\varepsilon\tau_0)\Delta_n\Pi_0 w(\tau_0,\varepsilon) + B(\varepsilon\tau_0)\Delta_n\Pi_0 u(\tau_0,\varepsilon))]_{2n-1} \\
&+ [\overline{\widetilde{\psi}}_{n-1}(\varepsilon\tau_0,\varepsilon)'(\{\varepsilon\widehat{\Pi}_{0(n-1)}f_w(\tau_0,\varepsilon)\}_{n-1}(\Delta_n\overline{w}(\varepsilon\tau_0,\varepsilon) + \Delta_n\Pi_0 w(\tau_0,\varepsilon)) \\
&\quad + \{\varepsilon\widehat{\Pi}_{0(n-1)}f_u(\tau_0,\varepsilon)\}_{n-1}(\Delta_n\overline{u}(\varepsilon\tau_0,\varepsilon) + \Delta_n\Pi_0 u(\tau_0,\varepsilon)))]_{2n-1} \\
&+ [\widetilde{\Pi}_{0(n-1)}\psi'(\varepsilon E_1 + E_2 + E_3)(\{\varepsilon f_w(\overline{\widetilde{\vartheta}}_{n-1}(\varepsilon\tau_0,\varepsilon) + \widetilde{\Pi}_{0(n-1)}\vartheta,\varepsilon\tau_0,\varepsilon)\}_{n-1}(\Delta_n\overline{w}(\varepsilon\tau_0,\varepsilon) \\
&\quad + \Delta_n\Pi_0 w(\tau_0,\varepsilon)) + \{\varepsilon f_u(\overline{\widetilde{\vartheta}}_{n-1}(\varepsilon\tau_0,\varepsilon) + \widetilde{\Pi}_{0(n-1)}\vartheta,\varepsilon\tau_0,\varepsilon)\}_{n-1}(\Delta_n\overline{u}(\varepsilon\tau_0,\varepsilon) \\
&\quad + \Delta_n\Pi_0 u(\tau_0,\varepsilon)))]_{2n-1})\,d\tau_0.
\end{aligned}
$$

In view of (37) and (39), we obtain from the last expression, omitting known terms, the following:

$$
\begin{aligned}
\int_0^{+\infty} &([(\Delta_n\overline{w}(\varepsilon\tau_0,\varepsilon) + \Delta_n\Pi_0 w(\tau_0,\varepsilon))'\{(E_1 + E_2 + \varepsilon E_3)\frac{d\widetilde{\Pi}_{0(n-1)}\psi(\tau_0,\varepsilon)}{d\tau_0}\}_{n-1}]_{2n-1} \\
&+ [\widetilde{\Pi}_{0(n-1)}\psi(\tau_0,\varepsilon)'(\varepsilon E_1 + E_2 + E_3)((E_1 + \varepsilon E_2 + \varepsilon^2 E_3)(\frac{d\overline{\widetilde{w}}_{n-1}(\varepsilon\tau_0,\varepsilon)}{dt} \\
&\quad + \frac{d\Delta_n\overline{w}(\varepsilon\tau_0,\varepsilon)}{dt}) - A(\varepsilon\tau_0)\overline{\widetilde{w}}_{n-1}(\varepsilon\tau_0,\varepsilon) - B(\varepsilon\tau_0)\widetilde{u}_{n-1}(\varepsilon\tau_0,\varepsilon) \\
&\quad - \{\varepsilon\overline{\widehat{f}}_{\vartheta(n-1)}(\varepsilon\tau_0,\varepsilon)\}_{n-1}\Delta_n\overline{\vartheta}(\varepsilon\tau_0,\varepsilon))]_{2n-1} \\
&+ [\widetilde{\Pi}_{0(n-1)}\psi(\tau_0,\varepsilon)'(\varepsilon E_1 + E_2 + E_3)((\frac{1}{\varepsilon}E_1 + E_2 + \varepsilon E_3)(\frac{d\widetilde{\Pi}_{0(n-1)}w(\tau_0,\varepsilon)}{d\tau_0} + \frac{d\Delta_n\Pi_0 w(\tau_0,\varepsilon)}{d\tau_0}) \\
&\quad - A(\varepsilon\tau_0)\widetilde{\Pi}_{0(n-1)}w(\tau_0,\varepsilon) - B(\varepsilon\tau_0)\widetilde{\Pi}_{0(n-1)}u(\tau_0,\varepsilon) - \{\varepsilon\widehat{\Pi}_{0(n-1)}f_\vartheta(\tau_0,\varepsilon)\}_{n-1}\Delta_n\overline{\vartheta}(\varepsilon\tau_0,\varepsilon) \\
&\quad - \{\varepsilon f_\vartheta(\overline{\widetilde{\vartheta}}_{n-1}(\varepsilon\tau_0,\varepsilon) + \widetilde{\Pi}_{0(n-1)}\vartheta(\tau_0,\varepsilon),\varepsilon\tau_0,\varepsilon)\}_{n-1}\Delta_n\Pi_0\vartheta(\tau_0,\varepsilon))]_{2n-1} \\
&+ [\overline{\widetilde{\psi}}_{n-1}(\varepsilon\tau_0,\varepsilon)'(\{\varepsilon\widehat{\Pi}_{0(n-1)}f_w(\tau_0,\varepsilon)\}_{n-1}(\Delta_n\overline{w}(\varepsilon\tau_0,\varepsilon) + \Delta_n\Pi_0 w(\tau_0,\varepsilon))
\end{aligned}
$$

$$+\{\varepsilon\widehat{\Pi}_{0(n-1)}f_u(\tau_0,\varepsilon)\}_{n-1}(\Delta_n\overline{u}(\varepsilon\tau_0,\varepsilon)+\Delta_n\Pi_0 u(\tau_0,\varepsilon)))]_{2n-1}$$

$$+[\widetilde{\Pi}_{0(n-1)}\psi(\tau_0,\varepsilon)'(\varepsilon E_1+E_2+E_3)(\{\varepsilon f_w(\widetilde{\overline{\vartheta}}_{n-1}(\varepsilon\tau_0,\varepsilon)$$

$$+\widetilde{\Pi}_{0(n-1)}\vartheta(\tau_0,\varepsilon),\varepsilon\tau_0,\varepsilon)\}_{n-1}(\Delta_n\overline{w}(\varepsilon\tau_0,\varepsilon)+\Delta_n\Pi_0 w(\tau_0,\varepsilon))+\{\varepsilon f_u(\widetilde{\overline{\vartheta}}_{n-1}(\varepsilon\tau_0,\varepsilon)$$

$$+\widetilde{\Pi}_{0(n-1)}\vartheta(\tau_0,\varepsilon),\varepsilon\tau_0,\varepsilon)\}_{n-1}(\Delta_n\overline{u}(\varepsilon\tau_0,\varepsilon)+\Delta_n\Pi_0 u(\tau_0,\varepsilon)))]_{2n-1})\,d\tau_0.$$

Integrating by parts in the last expression and dropping known terms, we obtain

$$-[(\Delta_n\overline{w}(0,\varepsilon)+\Delta_n\Pi_0 w(0,\varepsilon))'(E_1+E_2+\varepsilon E_3)\widetilde{\Pi}_{0(n-1)}\psi(0,\varepsilon)]_{2n-1}$$

$$+\int_0^{+\infty}[\widetilde{\overline{\psi}}_{n-1}(\varepsilon\tau_0,\varepsilon)'\{\varepsilon\widehat{\Pi}_{0(n-1)}f_\vartheta(\tau_0,\varepsilon)\}_{n-1}(\Delta_n\overline{\vartheta}(\varepsilon\tau_0,\varepsilon)+\Delta_n\Pi_0\vartheta(\tau_0,\varepsilon))]_{2n-1}\,d\tau_0.$$

Summing up the results, obtained from the transformed terms of the integral $\int_0^{+\infty}\Pi_{0(2n-1)}\mathbb{F}\,d\tau_0$, after dropping known terms, we have

$$-[\Delta_n\Pi_0 w(0,\varepsilon)'(\frac{1}{\varepsilon}E_1+E_2+\varepsilon E_3)\widetilde{\overline{\psi}}_{n-1}(0,\varepsilon)]_{2n-1}$$

$$-[(\Delta_n\overline{w}(0,\varepsilon)+\Delta_n\Pi_0 w(0,\varepsilon))'(E_1+E_2+\varepsilon E_3)\widetilde{\Pi}_{0(n-1)}\psi(0,\varepsilon)]_{2n-1}.$$

Performing similar transformations for $\int_{-\infty}^0 Q_{0(2n-1)}\mathbb{F}\,d\sigma_0$, we obtain the following result:

$$[\Delta_n Q_0 w(0,\varepsilon)'(\frac{1}{\varepsilon}E_1+E_2+\varepsilon E_3)\widetilde{\overline{\psi}}_{n-1}(T,\varepsilon)]_{2n-1}$$

$$+[(\Delta_n\overline{w}(T,\varepsilon)+\Delta_n Q_0 w(0,\varepsilon))'(E_1+E_2+\varepsilon E_3)\widetilde{Q}_{0(n-1)}\psi(0,\varepsilon)]_{2n-1}.$$

Furthermore, we apply the analogous transformations for the forth term of $J_{2n}$. The integral over the interval $[0,+\infty)$ of unknown terms of $\Pi_{1(2n-2)}\mathbb{F}(\tau_1)$ is presented as the sum

$$\sum_{i=1}^4 s_i+\int_0^{+\infty}(\Pi_{1(n-1)}w(\tau_1)'([W(\varepsilon^2\tau_1)\widetilde{\overline{w}}_{n-1}(\varepsilon^2\tau_1,\varepsilon)]_{n-1}+[\widehat{\overline{F}}_{w(n-2)}(\varepsilon^2\tau_1,\varepsilon)']_{n-2})$$

$$+\Pi_{1(n-1)}u(\tau_1)'([R(\varepsilon^2\tau_1)\widetilde{\overline{u}}_{n-1}(\varepsilon^2\tau_1,\varepsilon)]_{n-1}+[\widehat{\overline{F}}_{u(n-2)}(\varepsilon^2\tau_1,\varepsilon)']_{n-2}))\,d\tau_1$$

$$+\int_0^{+\infty}(\Pi_{1(n-1)}w(\tau_1)'([W(\varepsilon^2\tau_1)\widetilde{\Pi}_{0(n-1)}w(\varepsilon\tau_1,\varepsilon)]_{n-1}+[\widehat{\Pi}_{0(n-2)}F_w(\varepsilon\tau_1,\varepsilon)']_{n-2})$$

$$+\Pi_{1(n-1)}u(\tau_1)'([R(\varepsilon^2\tau_1)\widetilde{\Pi}_{0(n-1)}u(\varepsilon\tau_1,\varepsilon)]_{n-1}+[\widehat{\Pi}_{0(n-2)}F_u(\varepsilon\tau_1,\varepsilon)']_{n-2}))\,d\tau_1 \tag{41}$$

$$+\int_0^{+\infty}(\Pi_{1(n-1)}w(\tau_1)'(1/2W(0)\Pi_{1(n-1)}w(\tau_1)+[W(\varepsilon^2\tau_1)\widetilde{\Pi}_{1(n-2)}w(\tau_1,\varepsilon)]_{n-1}$$

$$+[\widehat{\Pi}_{1(n-2)}F_w(\tau_1,\varepsilon)']_{n-2}+\Pi_{1(n-1)}u(\tau_1)'(1/2R(0)\Pi_{1(n-1)}u(\tau_1)$$

$$+[R(\varepsilon^2\tau_1)\widetilde{\Pi}_{1(n-2)}u(\tau_1,\varepsilon)]_{n-1}+[\widehat{\Pi}_{1(n-2)}F_u(\tau_1,\varepsilon)']_{n-2}))\,d\tau_1,$$

where the expressions for $s_i$, $i=\overline{1,4}$, will be written later when they will be transformed.

Substituting $\vartheta(t,\varepsilon)$ from (35) into (2) and considering terms depending on $\tau_1$, we obtain the equation

$$(\frac{1}{\varepsilon^2}E_1+\frac{1}{\varepsilon}E_2+E_3)(\frac{d\widetilde{\Pi}_{1(n-1)}w(\tau_1,\varepsilon)}{d\tau_1}+\frac{d\Delta_n\Pi_1 w(\tau_1,\varepsilon)}{d\tau_1})$$

$$=A(\varepsilon^2\tau_1)(\widetilde{\Pi}_{1(n-1)}w(\tau_1,\varepsilon)+\Delta_n\Pi_1 w(\tau_1,\varepsilon))+B(\varepsilon^2\tau_1)(\widetilde{\Pi}_{1(n-1)}u(\tau_1,\varepsilon)$$

$$+\Delta_n\Pi_1 u(\tau_1,\varepsilon))+\varepsilon(f(\widetilde{\overline{\vartheta}}_{n-1}(\varepsilon^2\tau_1,\varepsilon)+\widetilde{\Pi}_{0(n-1)}\vartheta(\varepsilon\tau_1,\varepsilon)+\widetilde{\Pi}_{1(n-1)}\vartheta(\tau_1,\varepsilon) \tag{42}$$

$$+\Delta_n\overline{\vartheta}(\varepsilon^2\tau_1,\varepsilon)+\Delta_n\Pi_0\vartheta(\varepsilon\tau_1,\varepsilon)+\Delta_n\Pi_1\vartheta(\tau_1,\varepsilon),\varepsilon^2\tau_1,\varepsilon)$$

$$-f(\widetilde{\overline{\vartheta}}_{n-1}(\varepsilon^2\tau_1,\varepsilon)+\widetilde{\Pi}_{0(n-1)}\vartheta(\varepsilon\tau_1,\varepsilon)+\Delta_n\overline{\vartheta}(\varepsilon^2\tau_1,\varepsilon)+\Delta_n\Pi_0\vartheta(\varepsilon\tau_1,\varepsilon),\varepsilon^2\tau_1,\varepsilon)).$$

Using (36) with $k = n - 2$ and $t = \varepsilon^2 \tau_1$ in the expression

$$s_1 = \int_0^{+\infty} ([\Delta_n \Pi_1 w(\tau_1, \varepsilon)'(\{W(\varepsilon^2\tau_1)\widetilde{\overline{w}}_{n-2}(\varepsilon^2\tau_1, \varepsilon)\}_{n-2} + \{\varepsilon \widehat{\overline{F}}_{w(n-2)}(\varepsilon^2\tau_1, \varepsilon)'\}_{n-2})]_{2n-2}$$

$$+ [\Delta_n \Pi_1 u(\tau_1, \varepsilon)'(\{R(\varepsilon^2\tau_1)\widetilde{\overline{u}}_{n-2}(\varepsilon^2\tau_1, \varepsilon)\}_{n-2} + \{\varepsilon \widehat{\overline{F}}_{u(n-2)}(\varepsilon^2\tau_1, \varepsilon)'\}_{n-2})]_{2n-2}) \, d\tau_1,$$

we obtain

$$\int_0^{+\infty} ([\Delta_n \Pi_1 w(\tau_1, \varepsilon)'(E_1 + \varepsilon E_2 + \varepsilon^2 E_3) \frac{d\widetilde{\overline{\psi}}_{n-2}}{dt}(\varepsilon^2\tau_1, \varepsilon))]_{2n-2}$$

$$+ [\Delta_n \Pi_1 w(\tau_1, \varepsilon)'(\{A(\varepsilon^2\tau_1)'\widetilde{\overline{\psi}}_{n-2}(\varepsilon^2\tau_1, \varepsilon)\}_{n-2} + \{\varepsilon \widehat{\overline{f}}_{w(n-2)}(\varepsilon^2\tau_1, \varepsilon)'\widetilde{\overline{\psi}}_{n-2}(\varepsilon^2\tau_1, \varepsilon)\}_{n-2})]_{2n-2}$$

$$+ [\Delta_n \Pi_1 u(\tau_1, \varepsilon)'(\{B(\varepsilon^2\tau_1)'\widetilde{\overline{\psi}}_{n-2}(\varepsilon^2\tau_1, \varepsilon)\}_{n-2} + \{\varepsilon \widehat{\overline{f}}_{u(n-2)}(\varepsilon^2\tau_1, \varepsilon)'\widetilde{\overline{\psi}}_{n-2}(\varepsilon^2\tau_1, \varepsilon)\}_{n-2})]_{2n-2}) \, d\tau_1.$$

Then, applying (38) with $k = 2n - 2$, $l = n - 2$, and (42), we have

$$\int_0^{+\infty} ([\Delta_n \Pi_1 w(\tau_1, \varepsilon)'(E_1 + \varepsilon E_2 + \varepsilon^2 E_3) \frac{d\widetilde{\overline{\psi}}_{n-2}}{dt}(\varepsilon^2\tau_1, \varepsilon)]_{2n-2}$$

$$+ [\widetilde{\overline{\psi}}_{n-2}(\varepsilon^2\tau_1, \varepsilon)'(A(\varepsilon^2\tau_1)\Delta_n \Pi_1 w(\tau_1, \varepsilon) + B(\varepsilon^2\tau_1)\Delta_n \Pi_1 u(\tau_1, \varepsilon))]_{2n-2}$$

$$+ [\widetilde{\overline{\psi}}_{n-2}(\varepsilon^2\tau_1, \varepsilon)'(\varepsilon \widehat{\overline{f}}_{w(n-2)}(\varepsilon^2\tau_1, \varepsilon)\Delta_n \Pi_1 w(\tau_1, \varepsilon) + \varepsilon \widehat{\overline{f}}_{u(n-2)}(\varepsilon^2\tau_1, \varepsilon)\Delta_n \Pi_1 u(\tau_1, \varepsilon))]_{2n-2}) \, d\tau_1$$

$$= \int_0^{+\infty} ([\Delta_n \Pi_1 w(\tau_1, \varepsilon)'(E_1 + \varepsilon E_2 + \varepsilon^2 E_3) \frac{d\widetilde{\overline{\psi}}_{n-2}}{dt}(\varepsilon^2\tau_1, \varepsilon)]_{2n-2}$$

$$+ [\widetilde{\overline{\psi}}_{n-2}(\varepsilon^2\tau_1, \varepsilon)'((\frac{1}{\varepsilon^2}E_1 + \frac{1}{\varepsilon}E_2 + E_3)(\frac{d\widetilde{\Pi}_{1(n-1)}w(\tau_1, \varepsilon)}{d\tau_1} + \frac{d\Delta_n \Pi_1 w(\tau_1, \varepsilon)}{d\tau_1})$$

$$- A(\varepsilon^2\tau_1)\widetilde{\Pi}_{1(n-1)}w(\tau_1, \varepsilon) - B(\varepsilon^2\tau_1)\widetilde{\Pi}_{1(n-1)}u(\tau_1, \varepsilon) - \varepsilon(f(\widetilde{\overline{\vartheta}}_{n-1}(\varepsilon^2\tau_1, \varepsilon) + \widetilde{\Pi}_{0(n-1)}\vartheta(\varepsilon\tau_1, \varepsilon)$$

$$+ \widetilde{\Pi}_{1(n-1)}\vartheta(\tau_1, \varepsilon) + \Delta_n \overline{\vartheta}(\varepsilon^2\tau_1, \varepsilon) + \Delta_n \Pi_0 \vartheta(\varepsilon\tau_1, \varepsilon) + \Delta_n \Pi_1 \vartheta(\tau_1, \varepsilon), \varepsilon^2\tau_1, \varepsilon)$$

$$- f(\widetilde{\overline{\vartheta}}_{n-1}(\varepsilon^2\tau_1, \varepsilon) + \widetilde{\Pi}_{0(n-1)}\vartheta(\varepsilon\tau_1, \varepsilon) + \Delta_n \overline{\vartheta}(\varepsilon^2\tau_1, \varepsilon) + \Delta_n \Pi_0 \vartheta(\varepsilon\tau_1, \varepsilon), \varepsilon^2\tau_1, \varepsilon)))]_{2n-2}$$

$$+ [\widetilde{\overline{\psi}}_{n-2}(\varepsilon^2\tau_1, \varepsilon)'\varepsilon \widehat{\overline{f}}_{\vartheta(n-2)}(\varepsilon^2\tau_1, \varepsilon)\Delta_n \Pi_1 \vartheta(\tau_1, \varepsilon)]_{2n-2}) \, d\tau_1.$$

Integrating by parts in the last relation, using Remark 1, dropping known terms, we obtain

$$- [\Delta_n \Pi_1 w(0, \varepsilon)'(\frac{1}{\varepsilon^2}E_1 + \frac{1}{\varepsilon}E_2 + E_3)\widetilde{\overline{\psi}}_{n-2}(0, \varepsilon)]_{2n-2}$$

$$+ \int_0^{+\infty} ([\underbrace{\widetilde{\overline{\psi}}_{n-2}(\varepsilon^2\tau_1, \varepsilon)'(\frac{1}{\varepsilon^2}E_1 + \frac{1}{\varepsilon}E_2 + E_3)\frac{d\widetilde{\Pi}_{1(n-1)}w(\tau_1, \varepsilon)}{d\tau_1}]_{2n-2}}_{\{1\}}$$

$$+ [\widetilde{\overline{\psi}}_{n-2}(\varepsilon^2\tau_1, \varepsilon)'\varepsilon \widehat{\overline{f}}_{\vartheta(n-2)}(\varepsilon^2\tau_1, \varepsilon)\Delta_n \Pi_1 \vartheta(\tau_1, \varepsilon)]_{2n-2}$$

$$- [\widetilde{\overline{\psi}}_{n-2}(\varepsilon^2\tau_1, \varepsilon)'(\varepsilon\widehat{\Pi}_{1(n-2)}f_\vartheta(\tau_1, \varepsilon)(\Delta_n \overline{\vartheta}(\varepsilon^2\tau_1, \varepsilon) + \Delta_n \Pi_0 \vartheta(\varepsilon\tau_1, \varepsilon))$$

$$+ \varepsilon(\underbrace{\widehat{\overline{f}}_{\vartheta(n-2)}(\varepsilon^2\tau_1, \varepsilon)}_{\{1\}} + \underbrace{\widehat{\Pi}_{0(n-2)}f_\vartheta(\varepsilon\tau_1, \varepsilon)}_{\{2\}} + \widehat{\Pi}_{1(n-2)}f_\vartheta(\tau_1, \varepsilon))\varepsilon^{n-1}\Pi_{1(n-1)}\vartheta(\tau_1) \qquad (43)$$

$$+ \varepsilon f_\vartheta(\widetilde{\overline{\vartheta}}_{n-2}(\varepsilon^2\tau_1, \varepsilon) + \widetilde{\Pi}_{0(n-2)}\vartheta(\varepsilon\tau_1, \varepsilon) + \widetilde{\Pi}_{1(n-2)}\vartheta, \varepsilon^2\tau_1, \varepsilon)\Delta_n \Pi_1 \vartheta(\tau_1, \varepsilon))]_{2n-2}$$

$$- \underbrace{\Pi_{1(n-1)}w(\tau_1)'[A(\varepsilon^2\tau_1)'\widetilde{\overline{\psi}}_{n-2}(\varepsilon^2\tau_1, \varepsilon)]_{n-1}}_{\{1\}}$$

$$- \underbrace{\Pi_{1(n-1)}u(\tau_1)'[B(\varepsilon^2\tau_1)'\widetilde{\overline{\psi}}_{n-2}(\varepsilon^2\tau_1, \varepsilon)]_{n-1}}_{\{1\}}) \, d\tau_1.$$

Consider together the first integral in (41) and some terms with $\Pi_{1(n-1)}\vartheta(\tau_1,\varepsilon)$ in the transformed last expression for $s_1$, marked by $\{1\}$, namely, the expression of the form

$$\int_0^{+\infty}(\Pi_{1(n-1)}w(\tau_1)'([W(\varepsilon^2\tau_1)\widetilde{\overline{w}}_{n-1}(\varepsilon^2\tau_1,\varepsilon)]_{n-1}+[\widehat{\overline{F}}_{w(n-2)}(\varepsilon^2\tau_1,\varepsilon)']_{n-2}$$

$$-[A(\varepsilon^2\tau_1)'\widetilde{\overline{\psi}}_{n-2}(\varepsilon^2\tau_1,\varepsilon)]_{n-1}-[\widehat{\overline{f}}_{w(n-2)}(\varepsilon^2\tau_1,\varepsilon)'\widetilde{\overline{\psi}}_{n-2}(\varepsilon^2\tau_1,\varepsilon)]_{n-2})$$

$$+\Pi_{1(n-1)}u(\tau_1)'([R(\varepsilon^2\tau_1)\widetilde{\overline{u}}_{n-1}(\varepsilon^2\tau_1,\varepsilon)]_{n-1}+[\widehat{\overline{F}}_{u(n-2)}(\varepsilon^2\tau_1,\varepsilon)']_{n-2}$$

$$-[B(\varepsilon^2\tau_1)'\widetilde{\overline{\psi}}_{n-2}(\varepsilon^2\tau_1,\varepsilon)]_{n-1}-[\widehat{\overline{f}}_{u(n-2)}(\varepsilon^2\tau_1,\varepsilon)'\widetilde{\overline{\psi}}_{n-2}(\varepsilon^2\tau_1,\varepsilon)]_{n-2})$$

$$+[\widetilde{\overline{\psi}}_{n-2}(\varepsilon^2\tau_1,\varepsilon)'(\frac{1}{\varepsilon^2}E_1+\frac{1}{\varepsilon}E_2+E_3)\frac{d\widetilde{\Pi}_{1(n-1)}w(\tau_1,\varepsilon)}{d\tau_1}]_{2n-2})\,d\tau_1.$$

Transforming this expression with the help of (36) with $k=n-1$ and (10) at $j=n-1,n,n+1$ and omitting some known terms, we have

$$\int_0^{+\infty}(\Pi_{1(n-1)}w(\tau_1)'[E_1\frac{d\widetilde{\overline{\psi}}_{n-1}}{dt}(\varepsilon^2\tau_1,\varepsilon)+E_2\frac{d\widetilde{\overline{\psi}}_{n-2}}{dt}(\varepsilon^2\tau_1,\varepsilon)+E_3\frac{d\widetilde{\overline{\psi}}_{n-3}}{dt}(\varepsilon^2\tau_1,\varepsilon)]_{n-1}$$

$$+\overline{\psi}_{n-1}(0)'(E_1\frac{d\Pi_{1(n+1)}w(\tau_1)}{d\tau_1}+E_2\frac{d\Pi_{1n}w(\tau_1)}{d\tau_1}+E_3\frac{d\Pi_{1(n-1)}w(\tau_1)}{d\tau_1}$$

$$+[\widetilde{\overline{\psi}}_{n-2}(\varepsilon^2\tau_1,\varepsilon)'(\frac{1}{\varepsilon^2}E_1+\frac{1}{\varepsilon}E_2+E_3)\frac{d\widetilde{\Pi}_{1(n-1)}w(\tau_1,\varepsilon)}{d\tau_1})]_{2n-2})\,d\tau_1.$$

From here, using Remark 1, integrating by parts and omitting known terms, we obtain

$$-\overline{\psi}_{n-1}(0)'(E_1\Pi_{1(n+1)}w(0)+E_2\Pi_{1n}w(0)+E_3\Pi_{1(n-1)}w(0)).$$

Further changes concern the expression

$$s_2=\int_0^{+\infty}([\Delta_n\Pi_1w(\tau_1,\varepsilon)'(\{W(\varepsilon^2\tau_1)\widetilde{\Pi}_{0(n-2)}w(\varepsilon\tau_1,\varepsilon)\}_{n-2}$$

$$+\{\varepsilon\widehat{\Pi}_{0(n-2)}F_w(\varepsilon\tau_1,\varepsilon)\}_{n-2})]_{2n-2}+[\Delta_n\Pi_1u(\tau_1,\varepsilon)'(\{R(\varepsilon^2\tau_1,\varepsilon)\widetilde{\Pi}_{0(n-2)}u(\varepsilon\tau_1,\varepsilon)\}_{n-2}$$

$$+\{\varepsilon\widehat{\Pi}_{0(n-2)}F_u(\varepsilon\tau_1,\varepsilon)\}_{n-2})]_{2n-2})\,d\tau_1.$$

It will be transformed using (40) with $k=n-2$ and (38) in the following way:

$$\int_0^{+\infty}([\Delta_n\Pi_1w(\tau_1,\varepsilon)'(E_1+E_2+\varepsilon E_3)\frac{d\widetilde{\Pi}_{0(n-2)}\psi(\varepsilon\tau_1,\varepsilon)}{d\tau_0}]_{2n-2}$$

$$+[\widetilde{\Pi}_{0(n-2)}\psi(\varepsilon\tau_1,\varepsilon)'(\varepsilon E_1+E_2+E_3)(A(\varepsilon^2\tau_1)\Delta_n\Pi_1w(\tau_1,\varepsilon)+B(\varepsilon^2\tau_1)\Delta_n\Pi_1u(\tau_1,\varepsilon))]_{2n-2}$$

$$+[\Delta_n\Pi_1w(\tau_1,\varepsilon)'(\{\varepsilon\widehat{\Pi}_{0(n-2)}f_w(\varepsilon\tau_1,\varepsilon)'\widetilde{\overline{\psi}}_{n-2}(\varepsilon^2\tau_1,\varepsilon)\}_{n-2}$$

$$+\{\varepsilon f_w(\widetilde{\overline{\vartheta}}_{n-2}(\varepsilon^2\tau_1,\varepsilon)+\widetilde{\Pi}_{0(n-2)}\vartheta(\varepsilon\tau_1,\varepsilon),\varepsilon^2\tau_1,\varepsilon)'(\varepsilon E_1+E_2+E_3)\widetilde{\Pi}_{0(n-2)}\psi(\varepsilon\tau_1,\varepsilon)\}_{n-2})]_{2n-2}$$

$$+[\Delta_n\Pi_1u(\tau_1,\varepsilon)'(\{\varepsilon\widehat{\Pi}_{0(n-2)}f_u(\varepsilon\tau_1,\varepsilon)'\widetilde{\overline{\psi}}_{n-2}(\varepsilon^2\tau_1,\varepsilon)\}_{n-2}+\{\varepsilon f_u(\widetilde{\overline{\vartheta}}_{n-2}(\varepsilon^2\tau_1,\varepsilon)$$

$$+\widetilde{\Pi}_{0(n-2)}\vartheta(\varepsilon\tau_1,\varepsilon),\varepsilon^2\tau_1,\varepsilon)'(\varepsilon E_1+E_2+E_3)\widetilde{\Pi}_{0(n-2)}\psi(\varepsilon\tau_1,\varepsilon)\}_{n-2})]_{2n-2})\,d\tau_1.$$

From here, using (42) and omitting some known terms, we have

$$\int_0^{+\infty}([\Delta_n\Pi_1w(\tau_1,\varepsilon)'(E_1+E_2+\varepsilon E_3)\frac{d\widetilde{\Pi}_{0(n-2)}\psi(\varepsilon\tau_1,\varepsilon)}{d\tau_0}]_{2n-2}+[\widetilde{\Pi}_{0(n-2)}\psi(\varepsilon\tau_1,\varepsilon)'(\varepsilon E_1$$

$$+E_2+E_3)((\frac{1}{\varepsilon^2}E_1+\frac{1}{\varepsilon}E_2+E_3)(\frac{d\widetilde{\Pi}_{1(n-1)}w(\tau_1,\varepsilon)}{d\tau_1}+\frac{d\Delta_n\Pi_1w(\tau_1,\varepsilon)}{d\tau_1})$$

$$-A(\varepsilon^2\tau_1)\widetilde{\Pi}_{1(n-1)}w(\tau_1,\varepsilon) - B(\varepsilon^2\tau_1)\widetilde{\Pi}_{1(n-1)}u(\tau_1,\varepsilon) - \varepsilon\widehat{\Pi}_{1(n-2)}f_\vartheta(\tau_1,\varepsilon)(\varepsilon^{n-1}\overline{\vartheta}_{n-1}(\varepsilon^2\tau_1,\varepsilon)$$

$$+\varepsilon^{n-1}\Pi_{0(n-1)}\vartheta(\varepsilon\tau_1,\varepsilon) + \Delta_n\overline{\vartheta}(\varepsilon^2\tau_1,\varepsilon) + \Delta_n\Pi_0\vartheta(\varepsilon\tau_1,\varepsilon)) - \varepsilon f_\vartheta(\widetilde{\overline{\vartheta}}_{n-2}(\varepsilon^2\tau_1,\varepsilon)$$

$$+\widetilde{\Pi}_{0(n-2)}\vartheta(\varepsilon\tau_1,\varepsilon) + \widetilde{\Pi}_{1(n-2)}\vartheta(\tau_1,\varepsilon),\varepsilon^2\tau_1,\varepsilon)\}_{n-2})(\varepsilon^{n-1}\Pi_{1(n-1)}\vartheta(\tau_1)$$

$$+\Delta_n\Pi_1\vartheta(\tau_1,\varepsilon)))]_{2n-2} + [\Delta_n\Pi_1 w(\tau_1,\varepsilon)'(\{\varepsilon\widehat{\Pi}_{0(n-2)}f_w(\varepsilon\tau_1,\varepsilon)'\widetilde{\overline{\psi}}_{n-2}(\varepsilon^2\tau_1,\varepsilon)\}_{n-2}$$

$$+\{\varepsilon f_w(\widetilde{\overline{\vartheta}}_{n-2}(\varepsilon^2\tau_1,\varepsilon) + \widetilde{\Pi}_{0(n-2)}\vartheta(\varepsilon\tau_1,\varepsilon),\varepsilon^2\tau_1,\varepsilon)'(\varepsilon E_1 + E_2 + E_3)\widetilde{\Pi}_{0(n-2)}\psi(\varepsilon\tau_1,\varepsilon)\}_{n-2})]_{2n-2}$$

$$+[\Delta_n\Pi_1 u(\tau_1,\varepsilon)'(\{\varepsilon\widehat{\Pi}_{0(n-2)}f_u(\varepsilon\tau_1,\varepsilon)'\widetilde{\overline{\psi}}_{n-2}(\varepsilon^2\tau_1,\varepsilon)\}_{n-2} + \{\varepsilon f_u(\widetilde{\overline{\vartheta}}_{n-2}(\varepsilon^2\tau_1,\varepsilon)$$

$$+\widetilde{\Pi}_{0(n-2)}\vartheta(\varepsilon\tau_1,\varepsilon),\varepsilon^2\tau_1,\varepsilon)'(\varepsilon E_1 + E_2 + E_3)\widetilde{\Pi}_{0(n-2)}\psi(\varepsilon\tau_1,\varepsilon)\}_{n-2})]_{2n-2}\,d\tau_1.$$

Integrating by parts the first term in the last expression and dropping known terms, we obtain

$$-[\widetilde{\Pi}_{0(n-2)}\psi(0,\varepsilon)'(\frac{1}{\varepsilon}E_1 + \frac{1}{\varepsilon}E_2 + E_3)\Delta_n\Pi_1 w(0,\varepsilon)]_{2n-2}$$

$$+\int_0^{+\infty}(\underbrace{[\widetilde{\Pi}_{0(n-2)}\psi(\varepsilon\tau_1,\varepsilon)'(\frac{1}{\varepsilon}E_1 + \frac{1}{\varepsilon}E_2 + E_3)\frac{d\widetilde{\Pi}_{1(n-1)}w(\tau_1,\varepsilon)}{d\tau_1}]_{2n-2}}_{\{2\}}$$

$$-[\widetilde{\Pi}_{0(n-2)}\psi(\varepsilon\tau_1,\varepsilon)'(\varepsilon E_1 + E_2 + E_3)(\varepsilon\widehat{\Pi}_{1(n-2)}f_\vartheta(\tau_1,\varepsilon)(\Delta_n\overline{\vartheta}(\varepsilon^2\tau_1,\varepsilon) + \Delta_n\Pi_0\vartheta(\varepsilon\tau_1,\varepsilon))$$

$$+\varepsilon(\widehat{\Pi}_{1(n-2)}f_\vartheta(\tau_1,\varepsilon) + \underbrace{f_\vartheta(\widetilde{\overline{\vartheta}}_{n-2}(\varepsilon^2\tau_1,\varepsilon) + \widetilde{\Pi}_{0(n-2)}\vartheta(\varepsilon\tau_1,\varepsilon),\varepsilon^2\tau_1,\varepsilon))}_{\{2\}}\varepsilon^{n-1}\Pi_{1(n-1)}\vartheta(\tau_1)$$

$$+\varepsilon f_\vartheta(\widetilde{\overline{\vartheta}}_{n-2}(\varepsilon^2\tau_1,\varepsilon) + \widetilde{\Pi}_{0(n-2)}\vartheta(\varepsilon\tau_1,\varepsilon) + \widetilde{\Pi}_{1(n-2)}\vartheta(\tau_1,\varepsilon),\varepsilon^2\tau_1,\varepsilon)\Delta_n\Pi_1\vartheta(\tau_1,\varepsilon))]_{2n-2}$$

$$-\underbrace{\Pi_{1(n-1)}w(\tau_1)'[A(\varepsilon^2\tau_1)'(\varepsilon E_1 + E_2 + E_3)\widetilde{\Pi}_{0(n-2)}\psi(\varepsilon\tau_1,\varepsilon)]_{n-1}}_{\{2\}}$$

$$-\underbrace{\Pi_{1(n-1)}u(\tau_1)'[B(\varepsilon^2\tau_1)'(\varepsilon E_1 + E_2 + E_3)\widetilde{\Pi}_{0(n-2)}\psi(\varepsilon\tau_1,\varepsilon)]_{n-1}}_{\{2\}}$$

$$+[\Delta_n\Pi_1\vartheta(\tau_1,\varepsilon)'(\{\varepsilon\widehat{\Pi}_{0(n-2)}f_\vartheta(\varepsilon\tau_1,\varepsilon)'\widetilde{\overline{\psi}}_{n-2}(\varepsilon^2\tau_1,\varepsilon)\}_{n-2} + \{\varepsilon f_\vartheta(\widetilde{\overline{\vartheta}}_{n-2}(\varepsilon^2\tau_1,\varepsilon)$$

$$+\widetilde{\Pi}_{0(n-2)}\vartheta(\varepsilon\tau_1,\varepsilon),\varepsilon^2\tau_1,\varepsilon)'(\varepsilon E_1 + E_2 + E_3)\widetilde{\Pi}_{0(n-2)}\psi(\varepsilon\tau_1,\varepsilon)\}_{n-2})]_{2n-2}\,d\tau_1. \tag{44}$$

Consider together the second integral in (41) and some terms with $\Pi_{1(n-1)}\vartheta(\tau_1,\varepsilon)$ from (43) and (44), marked by $\{2\}$, namely the expression of the form

$$\int_0^{+\infty}(\Pi_{1(n-1)}w(\tau_1)'([W(\varepsilon^2\tau_1)\widetilde{\Pi}_{0(n-1)}w(\varepsilon\tau_1,\varepsilon)]_{n-1} + [\widehat{\Pi}_{0(n-2)}F_w(\varepsilon\tau_1,\varepsilon)']_{n-2}$$

$$-[A(\varepsilon^2\tau_1)'(\varepsilon E_1 + E_2 + E_3)\widetilde{\Pi}_{0(n-2)}\psi(\varepsilon\tau_1,\varepsilon)]_{n-1} - [\varepsilon\widehat{\Pi}_{0(n-2)}f_w(\varepsilon\tau_1,\varepsilon)'\widetilde{\overline{\psi}}_{n-2}(\varepsilon^2\tau_1,\varepsilon)]_{n-1}$$

$$-[\varepsilon f_w(\widetilde{\overline{\vartheta}}_{n-2}(\varepsilon^2\tau_1,\varepsilon) + \widetilde{\Pi}_{0(n-2)}\vartheta(\varepsilon\tau_1,\varepsilon),\varepsilon^2\tau_1,\varepsilon)'(\varepsilon E_1 + E_2 + E_3)\widetilde{\Pi}_{0(n-2)}\psi(\varepsilon\tau_1,\varepsilon)]_{n-1})$$

$$+\Pi_{1(n-1)}u(\tau_1)'([R(\varepsilon^2\tau_1)\widetilde{\Pi}_{0(n-1)}u(\varepsilon\tau_1,\varepsilon)]_{n-1} + [\widehat{\Pi}_{0(n-2)}F_u(\varepsilon\tau_1,\varepsilon)']_{n-2}$$

$$-[B(\varepsilon^2\tau_1)'(\varepsilon E_1 + E_2 + E_3)\widetilde{\Pi}_{0(n-2)}\psi(\varepsilon\tau_1,\varepsilon)]_{n-1} - [\varepsilon\widehat{\Pi}_{0(n-2)}f_u(\varepsilon\tau_1,\varepsilon)'\widetilde{\overline{\psi}}_{n-2}(\varepsilon^2\tau_1,\varepsilon)]_{n-1}$$

$$-[\varepsilon f_u(\widetilde{\overline{\vartheta}}_{n-2}(\varepsilon^2\tau_1,\varepsilon) + \widetilde{\Pi}_{0(n-2)}\vartheta(\varepsilon\tau_1,\varepsilon),\varepsilon^2\tau_1,\varepsilon)'(\varepsilon E_1 + E_2 + E_3)\widetilde{\Pi}_{0(n-2)}\psi(\varepsilon\tau_1,\varepsilon)]_{n-1})$$

$$+[\widetilde{\Pi}_{0(n-2)}\psi(\varepsilon\tau_1,\varepsilon)'(\frac{1}{\varepsilon}E_1 + \frac{1}{\varepsilon}E_2 + E_3)\frac{d\widetilde{\Pi}_{1(n-1)}w(\tau_1,\varepsilon)}{d\tau_1}]_{2n-2})\,d\tau_1.$$

We will transform this expression with the help of (40) with $k = n - 1$ and (10) with $j = n - 1, n$. Omitting known terms, we obtain

$$\int_0^{+\infty}([\widetilde{\Pi}_{0(n-2)}\psi(\varepsilon\tau_1,\varepsilon)'(\frac{1}{\varepsilon}E_1 + \frac{1}{\varepsilon}E_2 + E_3)\frac{d\widetilde{\Pi}_{1(n-1)}w(\tau_1,\varepsilon)}{d\tau_1}]_{2n-2}$$

$$+\Pi_{1(n-1)}w(\tau_1)'[(E_1 + E_2)\frac{d\widetilde{\Pi}_{0(n-1)}\psi}{d\tau_0}(\varepsilon\tau_1,\varepsilon) + E_3\frac{d\widetilde{\Pi}_{0(n-2)}\psi}{d\tau_0}(\varepsilon\tau_1,\varepsilon)]_{n-1}$$

$$+\Pi_{0(n-1)}\psi(0)'(E_2\frac{d\Pi_{1n}w(\tau_1)}{d\tau_1} + E_3\frac{d\Pi_{1(n-1)}w(\tau_1)}{d\tau_1}))\,d\tau_1.$$

Integrating by parts, using Remark 1 and omitting known terms, we have

$$-\Pi_{0(n-1)}\psi(0)'(E_2\Pi_{1n}w(0) + E_3\Pi_{1(n-1)}w(0)).$$

Multiplying Equations (24), (25) by $\varepsilon^j$, $j = \overline{0, n-2}$ and summing up the obtained results, we obtain the equalities

$$\{R(\varepsilon^2\tau_1)\widetilde{\Pi}_{1(n-2)}u(\tau_1,\varepsilon)\}_{n-2} + \{\varepsilon\widehat{\Pi}_{1(n-3)}F_u(\tau_1,\varepsilon)'\}_{n-2}$$

$$= \{B(\varepsilon^2\tau_1)'(\varepsilon^2 E_1 + \varepsilon E_2 + E_3)\widetilde{\Pi}_{1(n-2)}\psi(\tau_1,\varepsilon)\}_{n-2}$$

$$+\{\varepsilon\widehat{\Pi}_{1(n-3)}f_u(\tau_1,\varepsilon)'\overline{\widetilde{\psi}}_{n-3}(\varepsilon^2\tau_1,\varepsilon)\}_{n-2}$$

$$+\{\varepsilon\widehat{\Pi}_{1(n-3)}f_u(\tau_1,\varepsilon)'(\varepsilon E_1 + E_2 + E_3)\widetilde{\Pi}_{0(n-3)}\psi(\varepsilon\tau_1,\varepsilon)\}_{n-2}$$

$$+\{\varepsilon f_u(\overline{\widetilde{\vartheta}}_{n-3}(\varepsilon^2\tau_1,\varepsilon) + \widetilde{\Pi}_{0(n-3)}\vartheta(\varepsilon\tau_1,\varepsilon) + \widetilde{\Pi}_{1(n-3)}\vartheta(\tau_1,\varepsilon), \varepsilon^2\tau_1,\varepsilon)'(\varepsilon^2 E_1$$

$$+\varepsilon E_2 + E_3)\widetilde{\Pi}_{1(n-3)}\psi(\tau_1,\varepsilon)\}_{n-2}, \tag{45}$$

$$\{W(\varepsilon^2\tau_1)\widetilde{\Pi}_{1(n-2)}w(\tau_1,\varepsilon)\}_{n-2} + \{\varepsilon\widehat{\Pi}_{1(n-3)}F_w(\tau_1,\varepsilon)'\}_{n-2} = \frac{d\widetilde{\Pi}_{1(n-2)}\psi(\tau_1,\varepsilon)}{d\tau_1}$$

$$+\{A(\varepsilon^2\tau_1)'(\varepsilon^2 E_1 + \varepsilon E_2 + E_3)\widetilde{\Pi}_{1(n-2)}\psi(\tau_1,\varepsilon)\}_{n-2}$$

$$+\{\varepsilon\widehat{\Pi}_{1(n-3)}f_w(\tau_1,\varepsilon)'\overline{\widetilde{\psi}}_{n-3}(\varepsilon^2\tau_1,\varepsilon)\}_{n-2}$$

$$+\{\varepsilon\widehat{\Pi}_{1(n-3)}f_w(\tau_1,\varepsilon)'(\varepsilon E_1 + E_2 + E_3)\widetilde{\Pi}_{0(n-3)}\psi(\varepsilon\tau_1,\varepsilon)\}_{n-2}$$

$$+\{\varepsilon f_w(\overline{\widetilde{\vartheta}}_{n-3}(\varepsilon^2\tau_1,\varepsilon) + \widetilde{\Pi}_{0(n-3)}\vartheta(\varepsilon\tau_1,\varepsilon) + \widetilde{\Pi}_{1(n-3)}\vartheta(\tau_1,\varepsilon), \varepsilon^2\tau_1,\varepsilon)'(\varepsilon^2 E_1$$

$$+\varepsilon E_2 + E_3)\widetilde{\Pi}_{1(n-3)}\psi(\tau_1,\varepsilon)\}_{n-2}. \tag{46}$$

We will transform the expression

$$s_3 = \int_0^{+\infty}([(\Delta_n\overline{w}(\varepsilon^2\tau_1,\varepsilon) + \Delta_n\Pi_0 w(\varepsilon\tau_1,\varepsilon))'(\{W(\varepsilon^2\tau_1)\widetilde{\Pi}_{1(n-2)}w(\tau_1,\varepsilon)\}_{n-2}$$

$$+\{\varepsilon\widehat{\Pi}_{1(n-2)}F_w(\tau_1,\varepsilon)'\}_{n-2})]_{2n-2} + [(\Delta_n\overline{u}(\varepsilon^2\tau_1,\varepsilon)$$

$$+\Delta_n\Pi_0 u(\varepsilon\tau_1,\varepsilon))'(\{R(\varepsilon^2\tau_1)\widetilde{\Pi}_{1(n-2)}u(\tau_1,\varepsilon)\}_{n-2} + \{\varepsilon\widehat{\Pi}_{1(n-2)}F_u(\tau_1,\varepsilon)'\}_{n-2})]_{2n-2})\,d\tau_1.$$

Using (45), (46), (38) with $k = 2n - 2$, $l = n - 2$, (37), (39) and omitting known terms, we have

$$\int_0^{+\infty}([(\Delta_n\overline{w}(\varepsilon^2\tau_1,\varepsilon) + \Delta_n\Pi_0 w(\varepsilon\tau_1,\varepsilon))'\frac{d\widetilde{\Pi}_{1(n-2)}\psi(\tau_1,\varepsilon)}{d\tau_1}]_{2n-2}$$

$$+[\widetilde{\Pi}_{1(n-2)}\psi(\tau_1,\varepsilon)'(\varepsilon^2 E_1 + \varepsilon E_2 + E_3)((E_1 + \varepsilon E_2 + \varepsilon^2 E_3)(\frac{d\overline{\widetilde{w}}_{n-1}(\varepsilon^2\tau_1,\varepsilon)}{dt}$$

$$+\frac{d\Delta_n\overline{w}(\varepsilon^2\tau_1,\varepsilon)}{dt}) - A(\varepsilon^2\tau_1)\overline{\widetilde{w}}_{n-1}(\varepsilon^2\tau_1,\varepsilon) - B(\varepsilon^2\tau_1)\overline{\widetilde{u}}_{n-1}(\varepsilon^2\tau_1,\varepsilon)$$

$$-\varepsilon\overline{\widehat{f}}_{\vartheta(n-2)}(\varepsilon^2\tau_1,\varepsilon)(\varepsilon^{n-1}\overline{\vartheta}_{n-1}(\varepsilon^2\tau_1,\varepsilon) + \Delta_n\overline{\vartheta}(\varepsilon^2\tau_1,\varepsilon)))]_{2n-2}$$

$$+[\widetilde{\Pi}_{1(n-2)}\psi(\tau_1,\varepsilon)'(\varepsilon^2 E_1+\varepsilon E_2+E_3)((\frac{1}{\varepsilon}E_1+E_2+\varepsilon E_3)(\frac{d\widetilde{\Pi}_{0(n-1)}w(\varepsilon\tau_1,\varepsilon)}{d\tau_0}$$

$$+\frac{d\Delta_n\Pi_0 w(\varepsilon\tau_1,\varepsilon)}{d\tau_0})-A(\varepsilon^2\tau_1)\widetilde{\Pi}_{0(n-1)}w(\varepsilon\tau_1,\varepsilon)-B(\varepsilon^2\tau_1)\widetilde{\Pi}_{0(n-1)}u(\varepsilon\tau_1,\varepsilon)$$

$$-\varepsilon\widehat{\Pi}_{0(n-2)}f_\vartheta(\varepsilon\tau_1,\varepsilon)(\varepsilon^{n-1}\overline{\vartheta}_{n-1}(\varepsilon^2\tau_1,\varepsilon)+\Delta_n\overline{\vartheta}(\varepsilon^2\tau_1,\varepsilon))$$

$$-\varepsilon f_\vartheta(\overline{\widetilde{\vartheta}}_{n-2}(\varepsilon^2\tau_1,\varepsilon)+\widetilde{\Pi}_{0(n-2)}\vartheta(\varepsilon\tau_1,\varepsilon),\varepsilon^2\tau_1,\varepsilon)(\varepsilon^{n-1}\Pi_{0(n-1)}\vartheta(\varepsilon\tau_1,\varepsilon)+\Delta_n\Pi_0\vartheta(\varepsilon\tau_1,\varepsilon)))]_{2n-2}$$

$$+[(\Delta_n\overline{\vartheta}(\varepsilon^2\tau_1,\varepsilon)+\Delta_n\Pi_0\vartheta(\varepsilon\tau_1,\varepsilon))'(\{\varepsilon\widehat{\Pi}_{1(n-2)}f_\vartheta(\tau_1,\varepsilon)'\overline{\widetilde{\psi}}_{n-2}(\varepsilon^2\tau_1,\varepsilon)\}_{n-2}$$

$$+\{\varepsilon\widehat{\Pi}_{1(n-2)}f_\vartheta(\tau_1,\varepsilon)'(\varepsilon E_1+E_2+E_3)\widetilde{\Pi}_{0(n-2)}\psi(\varepsilon\tau_1,\varepsilon)\}_{n-2}+\{\varepsilon f_\vartheta(\overline{\widetilde{\vartheta}}_{n-2}(\varepsilon^2\tau_1,\varepsilon)$$

$$+\widetilde{\Pi}_{0(n-2)}\vartheta(\varepsilon\tau_1,\varepsilon)+\widetilde{\Pi}_{1(n-2)}\vartheta(\tau_1,\varepsilon),\varepsilon^2\tau_1,\varepsilon)'(\varepsilon^2 E_1+\varepsilon E_2$$

$$+E_3)\widetilde{\Pi}_{1(n-2)}\psi(\tau_1,\varepsilon)\}_{n-2})]_{2n-2})\,d\tau_1.$$

From here, applying the formula of integrating by parts, and omitting known terms, we obtain the unknown part from $s_3$

$$-[(\Delta_n\overline{w}(0,\varepsilon)+\Delta_n\Pi_0 w(0,\varepsilon))'\widetilde{\Pi}_{1(n-2)}\psi(0,\varepsilon)]_{2n-2}$$

$$-\int_0^{+\infty}([\widetilde{\Pi}_{1(n-2)}\psi(\tau_1,\varepsilon)'(\varepsilon^2 E_1+\varepsilon E_2+E_3)(\varepsilon\overline{\widehat{f}}_{\vartheta(n-2)}(\varepsilon^2\tau_1,\varepsilon)\Delta_n\overline{\vartheta}(\varepsilon^2\tau_1,\varepsilon)$$

$$+\varepsilon\widehat{\Pi}_{0(n-2)}f_\vartheta(\varepsilon\tau_1,\varepsilon)\Delta_n\overline{\vartheta}(\varepsilon^2\tau_1,\varepsilon)$$

$$+\varepsilon f_\vartheta(\overline{\widetilde{\vartheta}}_{n-2}(\varepsilon^2\tau_1,\varepsilon)+\widetilde{\Pi}_{0(n-2)}\vartheta(\varepsilon\tau_1,\varepsilon),\varepsilon^2\tau_1,\varepsilon)\Delta_n\Pi_0\vartheta(\varepsilon\tau_1,\varepsilon))]_{2n-2}$$

$$-[(\Delta_n\overline{\vartheta}(\varepsilon^2\tau_1,\varepsilon)+\Delta_n\Pi_0\vartheta(\varepsilon\tau_1,\varepsilon))'(\{\varepsilon\widehat{\Pi}_{1(n-2)}f_\vartheta(\tau_1,\varepsilon)'\overline{\widetilde{\psi}}_{n-2}(\varepsilon^2\tau_1,\varepsilon)\}_{n-2}$$

$$+\{\varepsilon\widehat{\Pi}_{1(n-2)}f_\vartheta(\tau_1,\varepsilon)'(\varepsilon E_1+E_2+E_3)\widetilde{\Pi}_{0(n-2)}\psi(\varepsilon\tau_1,\varepsilon)\}_{n-2}$$

$$+\{\varepsilon f_\vartheta(\overline{\widetilde{\vartheta}}_{n-2}(\varepsilon^2\tau_1,\varepsilon)+\widetilde{\Pi}_{0(n-2)}\vartheta(\varepsilon\tau_1,\varepsilon)$$

$$+\widetilde{\Pi}_{1(n-2)}\vartheta(\tau_1,\varepsilon),\varepsilon^2\tau_1,\varepsilon)'(\varepsilon^2 E_1+\varepsilon E_2+E_3)\widetilde{\Pi}_{1(n-2)}\psi(\tau_1,\varepsilon)\}_{n-2})]_{2n-2})\,d\tau_1.$$

Furthermore, applying the same algorithm, we will transform the expression

$$s_4=\int_0^{+\infty}([\Delta_n\Pi_1 w(\tau_1,\varepsilon)'(\{W(\varepsilon^2\tau_1)\widetilde{\Pi}_{1(n-2)}w(\tau_1,\varepsilon)\}_{n-2}+\{\varepsilon\widehat{\Pi}_{1(n-2)}F_w(\tau_1,\varepsilon)'\}_{n-2})]_{2n-2}$$

$$+[\Delta_n\Pi_1 u(\tau_1,\varepsilon)'(\{R(\varepsilon^2\tau_1)\widetilde{\Pi}_{1(n-2)}u(\tau_1,\varepsilon)\}_{n-2}+\{\varepsilon\widehat{\Pi}_{1(n-2)}F_u(\tau_1,\varepsilon)'\}_{n-2})]_{2n-2})\,d\tau_1.$$

Using (45), (46), (38) with $k=2n-2$, $l=n-2$, (42) and omitting known terms, we obtain

$$\int_0^{+\infty}([\Delta_n\Pi_1 w(\tau_1,\varepsilon)'\frac{d\widetilde{\Pi}_{1(n-2)}\psi(\tau_1,\varepsilon)}{d\tau_1}]_{2n-2}+[\widetilde{\Pi}_{1(n-2)}\psi(\tau_1,\varepsilon)'(\varepsilon^2 E_1+\varepsilon E_2+E_3)((\frac{1}{\varepsilon^2}E_1$$

$$+\frac{1}{\varepsilon}E_2+E_3)(\frac{d\widetilde{\Pi}_{1(n-1)}w(\tau_1,\varepsilon)}{d\tau_1}+\frac{d\Delta_n\Pi_1 w(\tau_1,\varepsilon)}{d\tau_1})-A(\varepsilon^2\tau_1)\widetilde{\Pi}_{1(n-1)}w(\tau_1,\varepsilon)$$

$$-B(\varepsilon^2\tau_1)\widetilde{\Pi}_{1(n-1)}u(\tau_1,\varepsilon)-\varepsilon\widehat{\Pi}_{1(n-2)}f_\vartheta(\tau_1,\varepsilon)(\Delta_n\overline{\vartheta}(\varepsilon^2\tau_1,\varepsilon)+\Delta_n\Pi_0\vartheta(\varepsilon\tau_1,\varepsilon))$$

$$-\varepsilon f_\vartheta(\overline{\widetilde{\vartheta}}_{n-2}(\varepsilon^2\tau_1,\varepsilon)+\widetilde{\Pi}_{0(n-2)}\vartheta(\varepsilon\tau_1,\varepsilon)+\widetilde{\Pi}_{1(n-2)}(\tau_1,\varepsilon),\varepsilon^2\tau_1,\varepsilon)(\varepsilon^{n-1}\Pi_{1(n-1)}\vartheta(\tau_1)$$

$$+\Delta_n\Pi_1\vartheta(\tau_1,\varepsilon)))]_{2n-2}+[\,\Delta_n\Pi_1\vartheta(\tau_1,\varepsilon)'(\{\varepsilon\widehat{\Pi}_{1(n-2)}f_\vartheta(\tau_1,\varepsilon)'\overline{\widetilde{\psi}}_{n-2}(\varepsilon^2\tau_1,\varepsilon)\}_{n-2}$$

$$+\{\varepsilon\widehat{\Pi}_{1(n-2)}f_\vartheta(\tau_1,\varepsilon)'(\varepsilon E_1+E_2+E_3)\widetilde{\Pi}_{0(n-2)}\psi(\varepsilon\tau_1,\varepsilon)\}_{n-2}+\{\varepsilon f_\vartheta(\overline{\widetilde{\vartheta}}_{n-2}(\varepsilon^2\tau_1,\varepsilon)$$

$$+\widetilde{\Pi}_{0(n-2)}\vartheta(\varepsilon\tau_1,\varepsilon)+\widetilde{\Pi}_{1(n-2)}\vartheta(\tau_1,\varepsilon),\varepsilon^2\tau_1,\varepsilon)'(\varepsilon^2 E_1+\varepsilon E_2$$

$$+E_3)\widetilde{\Pi}_{1(n-2)}\psi(\tau_1,\varepsilon)\}_{n-2})]_{2n-2})\,d\tau_1.$$

Due to formula of integrating by parts, after omitting known terms, we obtain the following:

$$-[\Delta_n\Pi_1 w(0,\varepsilon)'\widetilde{\Pi}_{1(n-2)}\psi(0,\varepsilon)]_{2n-2}$$

$$+\int_0^{+\infty}(\Pi_{1(n-1)}w(\tau_1)'(-[A(\varepsilon^2\tau_1)'(\varepsilon^2 E_1+\varepsilon E_2+E_3)\widetilde{\Pi}_{1(n-2)}\psi(\tau_1,\varepsilon)]_{n-1}$$

$$-[\varepsilon f_w(\overline{\widetilde{\vartheta}}_{n-2}(\varepsilon^2\tau_1,\varepsilon)+\widetilde{\Pi}_{0(n-2)}\vartheta(\varepsilon\tau_1,\varepsilon)+\widetilde{\Pi}_{1(n-2)}\vartheta(\tau_1,\varepsilon),\varepsilon^2\tau_1,\varepsilon)'(\varepsilon^2 E_1+\varepsilon E_2$$

$$+E_3)\widetilde{\Pi}_{1(n-2)}\psi(\tau_1,\varepsilon)]_{n-1})-\Pi_{1(n-1)}u(\tau_1)'([B(\varepsilon^2\tau_1)'(\varepsilon^2 E_1+\varepsilon E_2+E_3)\widetilde{\Pi}_{1(n-2)}\psi(\tau_1,\varepsilon)]_{n-1}$$

$$-[\varepsilon f_u(\overline{\widetilde{\vartheta}}_{n-2}(\varepsilon^2\tau_1,\varepsilon)+\widetilde{\Pi}_{0(n-2)}\vartheta(\varepsilon\tau_1,\varepsilon)+\widetilde{\Pi}_{1(n-2)}\vartheta(\tau_1,\varepsilon),\varepsilon^2\tau_1,\varepsilon)'(\varepsilon^2 E_1$$

$$+\varepsilon E_2+E_3)\widetilde{\Pi}_{1(n-2)}\psi(\tau_1,\varepsilon)]_{n-1})-[\widetilde{\Pi}_{1(n-2)}\psi(\tau_1,\varepsilon)'(\varepsilon^2 E_1+\varepsilon E_2$$

$$+E_3)(\varepsilon\widehat{\Pi}_{1(n-2)}f_\vartheta(\tau_1,\varepsilon)(\Delta_n\overline{\vartheta}(\varepsilon^2\tau_1,\varepsilon)+\Delta_n\Pi_0\vartheta(\varepsilon\tau_1,\varepsilon))$$

$$+\varepsilon f_\vartheta(\overline{\widetilde{\vartheta}}_{n-2}(\varepsilon^2\tau_1,\varepsilon)+\widetilde{\Pi}_{0(n-2)}\vartheta(\varepsilon\tau_1,\varepsilon)+\widetilde{\Pi}_{1(n-2)}\vartheta(\tau_1,\varepsilon),\varepsilon^2\tau_1,\varepsilon)\Delta_n\Pi_1\vartheta(\tau_1,\varepsilon))]_{2n-2}$$

$$+[\Delta_n\Pi_1\vartheta(\tau_1,\varepsilon)'(\{\varepsilon\widehat{\Pi}_{1(n-2)}f_\vartheta(\tau_1,\varepsilon)'\overline{\widetilde{\psi}}_{n-2}(\varepsilon^2\tau_1,\varepsilon)\}_{n-2}$$

$$+\{\varepsilon\widehat{\Pi}_{1(n-2)}f_\vartheta(\tau_1,\varepsilon)'(\varepsilon E_1+E_2+E_3)\widetilde{\Pi}_{0(n-2)}\psi(\varepsilon\tau_1,\varepsilon)\}_{n-2}+\{\varepsilon f_\vartheta(\overline{\widetilde{\vartheta}}_{n-2}(\varepsilon^2\tau_1,\varepsilon)$$

$$+\widetilde{\Pi}_{0(n-2)}\vartheta(\varepsilon\tau_1,\varepsilon)+\widetilde{\Pi}_{1(n-2)}\vartheta(\tau_1,\varepsilon),\varepsilon^2\tau_1,\varepsilon)'(\varepsilon^2 E_1+\varepsilon E_2$$

$$+E_3)\widetilde{\Pi}_{1(n-2)}\psi(\tau_1,\varepsilon)\}_{n-2})]_{2n-2})\,d\tau_1.$$

Summing up the obtained terms of transformed expressions and considering separately four groups of terms, depending on $\Delta_n\overline{\vartheta}$, $\Delta_n\Pi_0\vartheta$, $\Delta_n\Pi_1\vartheta$, and without these variables, we can write out the transformed forth term of $J_{2n}$ in the following form:

$$-[\Delta_n\Pi_1 w(0,\varepsilon)'(\frac{1}{\varepsilon}E_1+E_2+\varepsilon E_3)\overline{\widetilde{\psi}}_{n-1}(0,\varepsilon)]_{2n-1}-\Pi_{1(n-1)}w(0)'E_3(\overline{\psi}_{n-1}(0)$$

$$+\Pi_{0(n-1)}\psi(0))-[\Delta_n\Pi_1 w(0,\varepsilon)'(E_1+E_2+\varepsilon E_3)\widetilde{\Pi}_{0(n-1)}\psi(0,\varepsilon)]_{2n-1}$$

$$-[(\Delta_n\overline{w}(0,\varepsilon)+\Delta_n\Pi_0 w(0,\varepsilon)+\Delta_n\Pi_1 w(0,\varepsilon))'\widetilde{\Pi}_{1(n-2)}\psi(0,\varepsilon)]_{2n-2}+\Pi_{1(n-1)}J.$$

Transforming the fifth term in $J_{2n}$ in the same way, we write the final result as

$$[\Delta_n Q_1 w(0,\varepsilon)'(\frac{1}{\varepsilon}E_1+E_2+\varepsilon E_3)\overline{\widetilde{\psi}}_{n-1}(T,\varepsilon)]_{2n-1}$$

$$+[\Delta_n Q_1 w(0,\varepsilon)'(E_1+E_2+\varepsilon E_3)\widetilde{Q}_{0(n-1)}\psi(0,\varepsilon)]_{2n-1}$$

$$+[(\Delta_n\overline{w}(T,\varepsilon)+\Delta_n Q_0 w(0,\varepsilon)+\Delta_n Q_1 w(0,\varepsilon))'\widetilde{Q}_{1(n-2)}\psi(0,\varepsilon)]_{2n-2}+Q_{1(n-1)}J.$$

Substituting $w(t,\varepsilon)$ from (35) in (3), we obtain the relation

$$\overline{\widetilde{w}}_{j-1}(0,\varepsilon)+\Delta_j\overline{w}(0,\varepsilon)+\sum_{i=0}^{1}(\widetilde{\Pi}_{i(j-1)}w(0,\varepsilon)+\Delta_j\Pi_i w(0,\varepsilon)$$

$$+\widetilde{Q}_{i(j-1)}w(-T/\varepsilon^{i+1},\varepsilon)+\Delta_j Q_i w(-T/\varepsilon^{i+1},\varepsilon))=w^0. \tag{47}$$

Summing up (20) and the second relations in (23), (26), we obtain the equality

$$\overline{\psi}_j(T)+E_1(Q_{0(j-1)}\psi(0)+Q_{1(j-2)}\psi(0))$$

$$+E_2(Q_{0j}\psi(0)+Q_{1(j-1)}\psi(0))+E_3(Q_{0j}\psi(0)+Q_{1j}\psi(0))=0.$$

Multiplying this equation by $\varepsilon^j$, $j=\overline{0,n-1}$, and summing up the obtained results, we have

$$\overline{\widetilde{\psi}}_{n-1}(T)+\varepsilon E_1(\widetilde{Q}_{0(n-2)}\psi(0)+\varepsilon\widetilde{Q}_{1(n-3)}\psi(0))$$

$$+E_2(\widetilde{Q}_{0(n-1)}\psi(0)+\varepsilon\widetilde{Q}_{1(n-2)}\psi(0))+E_3(\widetilde{Q}_{0(n-1)}\psi(0)+\widetilde{Q}_{1(n-1)}\psi(0))=0. \tag{48}$$

Summing up the remaining parts of the transformed terms for $J_{2n}$, applying (47), (48) to the non-integrand terms, taking into account Remark 1 and omitting known terms, we finally have $\bar{J}_n + \Pi_{1(n-1)}J + Q_{1(n-1)}J$, which proves Theorem 1 for $J_{2n}$.

In addition to the previous assumption on solvability of the problems $\bar{P}_j$, $\Pi_{0j}P$, $Q_{0j}P$ and $\Pi_{1(j-1)}P$, $Q_{1(j-1)}P$, $j = \overline{0, n-1}$, we will assume that the problems $\bar{P}_n$, $\Pi_{1(n-1)}P$, $Q_{1(n-1)}P$ have been solved.

Let us consider the coefficient $J_{2n+1}$ having the form (34) with $k = 2n + 1$.

We transform separately the terms of $J_{2n+1}$ using the previous algorithm for transforming similar terms in $J_{2n}$. Summing up the obtained expressions for five terms and dropping known terms, we obtain the sum of the performance indices $\Pi_{0n}J + Q_{0n}J$.

Thus, Theorem 1 is completely proved.  □

## 5. Asymptotic Estimates

Suppose that the problems $\bar{P}_j$, $\Pi_{ij}P$, $Q_{ij}P$, $i = 0, 1$, $j = \overline{0, n}$, have been solved. We will prove asymptotic estimates of the proximity between the asymptotic solution obtained by the direct scheme method $\tilde{\vartheta}_n(t, \varepsilon) = \sum_{j=0}^{n} \varepsilon^j (\bar{\vartheta}_j(t) + \sum_{i=0}^{1} \Pi_{ij}\vartheta(\tau_i) + Q_{ij}\vartheta(\sigma_i))$ and the exact solution of the problem $P_\varepsilon$.

We will use here the notation for asymptotics remainder terms

$$r_n w = w - \tilde{w}_n = (r_n x', r_n y', r_n z')', r_n u = u - \tilde{u}_n, r_n \vartheta = \vartheta - \tilde{\vartheta}_n = (r_n w', r_n u')',$$

$$r_n \varphi = \varphi - \tilde{\varphi}_n = (r_n \zeta', r_n \eta', r_n \theta')', \quad \tilde{X}_n = \begin{pmatrix} \tilde{x}_n \\ \tilde{\zeta}_n \end{pmatrix}, \tag{49}$$

$$\tilde{Y}_n = \begin{pmatrix} \tilde{y}_n \\ \tilde{\eta}_n \end{pmatrix}, \ \tilde{Z}_n = \begin{pmatrix} \tilde{z}_n \\ \tilde{\theta}_n \end{pmatrix}, \ r_n X = \begin{pmatrix} r_n x \\ r_n \zeta \end{pmatrix}, \ r_n Y = \begin{pmatrix} r_n y \\ r_n \eta \end{pmatrix}, \ r_n Z = \begin{pmatrix} r_n z \\ r_n \theta \end{pmatrix}.$$

In comparison with the notation in the previous section, we have, e.g., $r_n u = \Delta_{n+1} u$ and so on.

Since the matrix $R(t)$ is positive definite, we obtain from (27) the following relation

$$u(t, \varepsilon) = R(t)^{-1} B(t)' \varphi + \varepsilon R(t)^{-1} (f_u(\vartheta, t, \varepsilon)' \varphi - F_u(\vartheta, t, \varepsilon)').$$

Taking into account this equality and substituting the expressions for $\vartheta(t, \varepsilon)$, $\varphi(t, \varepsilon)$ from (49) into (2), (27) and (28), we obtain the equations for the remainders

$$r_n u = \overset{(1)}{\mathcal{A}}(t) r_n X + \overset{(1)}{\mathcal{B}}(t) r_n Y + \overset{(1)}{\mathcal{C}}(t) r_n Z + \overset{(1)}{g}(r_n \vartheta, r_n \varphi, t, \varepsilon), \tag{50}$$

$$\frac{d r_n X}{dt} = \overset{(2)}{\mathcal{A}}(t) r_n X + \overset{(2)}{\mathcal{B}}(t) r_n Y + \overset{(2)}{\mathcal{C}}(t) r_n Z + \overset{(2)}{g}(r_n \vartheta, r_n \varphi, t, \varepsilon), \tag{51}$$

$$\varepsilon \frac{d r_n Y}{dt} = \overset{(3)}{\mathcal{A}}(t) r_n X + \overset{(3)}{\mathcal{B}}(t) r_n Y + \overset{(3)}{\mathcal{C}}(t) r_n Z + \overset{(3)}{g}(r_n \vartheta, r_n \varphi, t, \varepsilon), \tag{52}$$

$$\varepsilon^2 \frac{d r_n Z}{dt} = \overset{(4)}{\mathcal{A}}(t) r_n X + \overset{(4)}{\mathcal{B}}(t) r_n Y + \overset{(4)}{\mathcal{C}}(t) r_n Z + \overset{(4)}{g}(r_n \vartheta, r_n \varphi, t, \varepsilon), \tag{53}$$

where

$$\overset{(1)}{\mathcal{A}} = (0 \ \ R^{-1}B'), \ \overset{(1)}{\mathcal{B}} = (0 \ \ R^{-1}B'), \ \overset{(1)}{\mathcal{C}} = (0 \ \ R^{-1}B'),$$

$$\overset{(2)}{\mathcal{A}} = \begin{pmatrix} A_{11} & S_{11} \\ W_{11} & -A'_{11} \end{pmatrix}, \ \overset{(2)}{\mathcal{B}} = \begin{pmatrix} A_{12} & S_{12} \\ W_{12} & -A'_{21} \end{pmatrix}, \ \overset{(2)}{\mathcal{C}} = \begin{pmatrix} A_{13} & S_{13} \\ W_{13} & -A'_{31} \end{pmatrix},$$

$$\overset{(3)}{\mathcal{A}} = \begin{pmatrix} A_{21} & S'_{12} \\ W'_{12} & -A'_{12} \end{pmatrix}, \ \overset{(3)}{\mathcal{B}} = \begin{pmatrix} A_{22} & S_{22} \\ W_{22} & -A'_{22} \end{pmatrix}, \ \overset{(3)}{\mathcal{C}} = \begin{pmatrix} A_{23} & S_{23} \\ W_{23} & -A'_{32} \end{pmatrix},$$

$$\overset{(4)}{\mathcal{A}} = \begin{pmatrix} A_{31} & S'_{13} \\ W'_{13} & -A'_{13} \end{pmatrix}, \overset{(4)}{\mathcal{B}} = \begin{pmatrix} A_{32} & S'_{23} \\ W'_{23} & -A'_{23} \end{pmatrix}, \overset{(4)}{\mathcal{C}} = \begin{pmatrix} A_{33} & S_{33} \\ W_{33} & -A'_{33} \end{pmatrix},$$

$$\overset{(1)}{g}(r_n \vartheta, r_n \varphi, t, \varepsilon) = R(t)^{-1} B(t)' \widetilde{\varphi}_n + \widetilde{u}_n$$
$$+ \varepsilon R(t)^{-1}(f_u(\widetilde{\vartheta}_n + r_n \vartheta, t, \varepsilon)'(\widetilde{\varphi}_n + r_n \varphi) - F_u(\widetilde{\vartheta}_n + r_n \vartheta, t, \varepsilon)'),$$

$$\overset{(2)}{g}(r_n \vartheta, r_n \varphi, t, \varepsilon) = \overset{(2)}{\mathcal{A}}(t) \widetilde{X}_n + \overset{(2)}{\mathcal{B}}(t) \widetilde{Y}_n + \overset{(2)}{\mathcal{C}}(t) \widetilde{Z}_n$$
$$- d\widetilde{X}_n / dt + \varepsilon \overset{(2)}{h}(\widetilde{\vartheta}_n + r_n \vartheta, \widetilde{\varphi}_n + r_n \varphi, t, \varepsilon),$$

$$\overset{(3)}{g}(r_n \vartheta, r_n \varphi, t, \varepsilon) = \overset{(3)}{\mathcal{A}}(t) \widetilde{X}_n + \overset{(3)}{\mathcal{B}}(t) \widetilde{Y}_n + \overset{(3)}{\mathcal{C}}(t) \widetilde{Z}_n$$
$$- \varepsilon d\widetilde{Y}_n / dt + \varepsilon \overset{(3)}{h}(\widetilde{\vartheta}_n + r_n \vartheta, \widetilde{\varphi}_n + r_n \varphi, t, \varepsilon),$$

$$\overset{(4)}{g}(r_n \vartheta, r_n \varphi, t, \varepsilon) = \overset{(4)}{\mathcal{A}}(t) \widetilde{X}_n + \overset{(4)}{\mathcal{B}}(t) \widetilde{Y}_n + \overset{(4)}{\mathcal{C}}(t) \widetilde{Z}_n$$
$$- \varepsilon^2 d\widetilde{Z}_n / dt + \varepsilon \overset{(4)}{h}(\widetilde{\vartheta}_n + r_n \vartheta, \widetilde{\varphi}_n + r_n \varphi, t, \varepsilon),$$

$$\overset{(2)}{h}(\widetilde{\vartheta}_n + r_n \vartheta, \widetilde{\varphi}_n + r_n \varphi, t, \varepsilon) = ((\overset{(1)}{B}(t)R(t)^{-1}(f_u(\widetilde{\vartheta}_n + r_n \vartheta, t, \varepsilon)'(\widetilde{\varphi}_n + r_n \varphi) - F_u(\widetilde{\vartheta}_n$$
$$+ r_n \vartheta, t, \varepsilon)') + \overset{(1)}{f}(\widetilde{\vartheta}_n + r_n \vartheta, t, \varepsilon))', (F_x(\widetilde{\vartheta}_n + r_n \vartheta, t, \varepsilon)' - \overset{(1)}{f_x}(\widetilde{\vartheta}_n + r_n \vartheta, t, \varepsilon)'(\widetilde{\varphi}_n + r_n \varphi))')',$$

$$\overset{(3)}{h}(\widetilde{\vartheta}_n + r_n \vartheta, \widetilde{\varphi}_n + r_n \varphi, t, \varepsilon) = ((\overset{(2)}{B}(t)R(t)^{-1}(f_u(\widetilde{\vartheta}_n + r_n \vartheta, t, \varepsilon)'(\widetilde{\varphi}_n + r_n \varphi) - F_u(\widetilde{\vartheta}_n$$
$$+ r_n \vartheta, t, \varepsilon)') + \overset{(2)}{f}(\widetilde{\vartheta}_n + r_n \vartheta, t, \varepsilon))', (F_y(\widetilde{\vartheta}_n + r_n \vartheta, t, \varepsilon)' - \overset{(2)}{f_y}(\widetilde{\vartheta}_n + r_n \vartheta, t, \varepsilon)'(\widetilde{\varphi}_n + r_n \varphi))')',$$

$$\overset{(4)}{h}(\widetilde{\vartheta}_n + r_n \vartheta, \widetilde{\varphi}_n + r_n \varphi, t, \varepsilon) = ((\overset{(3)}{B}(t)R(t)^{-1}(f_u(\widetilde{\vartheta}_n + r_n \vartheta, t, \varepsilon)'(\widetilde{\varphi}_n + r_n \varphi) - F_u(\widetilde{\vartheta}_n$$
$$+ r_n \vartheta, t, \varepsilon)') + \overset{(3)}{f}(\widetilde{\vartheta}_n + r_n \vartheta, t, \varepsilon))', (F_z(\widetilde{\vartheta}_n + r_n \vartheta, t, \varepsilon)' - \overset{(3)}{f_z}(\widetilde{\vartheta}_n + r_n \vartheta, t, \varepsilon)'(\widetilde{\varphi}_n + r_n \varphi))')'.$$

For brevity, the arguments $t, \varepsilon$ are dropped in some of the last relations.

In view of the algorithm of asymptotics construction, namely Equalities (12)–(17) and (33), we obtain the boundary conditions

$$\begin{aligned} r_n w(0, \varepsilon) &= -\widetilde{Q}_{0n} w(-T/\varepsilon, \varepsilon) - \widetilde{Q}_{1n} w(-T/\varepsilon^2, \varepsilon), \\ r_n \varphi(T, \varepsilon) &= -\widetilde{\Pi}_{0n} \varphi(T/\varepsilon, \varepsilon) - \widetilde{\Pi}_{1n} \varphi(T/\varepsilon^2, \varepsilon). \end{aligned} \tag{54}$$

Using variables' changes,

$$\rho_n w(t, \varepsilon) = r_n w(t, \varepsilon) - r_n w(0, \varepsilon), \quad \rho_n \varphi(t, \varepsilon) = r_n \varphi(t, \varepsilon) - r_n \varphi(T, \varepsilon) \tag{55}$$

and the notation $\rho_n v(t, \varepsilon) = (r_n u', \rho_n w', \rho_n \varphi')'$, system (50)–(54) can be written as

$$r_n u = \overset{(1)}{\mathcal{A}}(t) \rho_n X + \overset{(1)}{\mathcal{B}}(t) \rho_n Y + \overset{(1)}{\mathcal{C}}(t) \rho_n Z + \overset{(1)}{\chi}(\rho_n v, t, \varepsilon), \tag{56}$$

$$\frac{d\rho_n X}{dt} = \overset{(2)}{\mathcal{A}}(t)\rho_n X + \overset{(2)}{\mathcal{B}}(t)\rho_n Y + \overset{(2)}{\mathcal{C}}(t)\rho_n Z + \overset{(2)}{\chi}(\rho_n v, t, \varepsilon), \tag{57}$$

$$\varepsilon\frac{d\rho_n Y}{dt} = \overset{(3)}{\mathcal{A}}(t)\rho_n X + \overset{(3)}{\mathcal{B}}(t)\rho_n Y + \overset{(3)}{\mathcal{C}}(t)\rho_n Z + \overset{(3)}{\chi}(\rho_n v, t, \varepsilon), \tag{58}$$

$$\varepsilon^2\frac{d\rho_n Z}{dt} = \overset{(4)}{\mathcal{A}}(t)\rho_n X + \overset{(4)}{\mathcal{B}}(t)\rho_n Y + \overset{(4)}{\mathcal{C}}(t)\rho_n Z + \overset{(4)}{\chi}(\rho_n v, t, \varepsilon), \tag{59}$$

$$\rho_n w(0, \varepsilon) = 0, \ \rho_n \varphi(T, \varepsilon) = 0, \tag{60}$$

where

$$\rho_n X = \begin{pmatrix} \rho_n x \\ \rho_n \zeta \end{pmatrix}, \ \rho_n Y = \begin{pmatrix} \rho_n y \\ \rho_n \eta \end{pmatrix}, \ \rho_n Z = \begin{pmatrix} \rho_n z \\ \rho_n \theta \end{pmatrix},$$

$$\overset{(i)}{\chi}(\rho_n v, t, \varepsilon) = \overset{(i)}{g}(r_n u(t, \varepsilon), \rho_n w(t, \varepsilon) + r_n w(0, \varepsilon), \rho_n \varphi(t, \varepsilon) + r_n \varphi(T, \varepsilon), t, \varepsilon) +$$

$$+\overset{(i)}{\mathcal{A}}(t)\begin{pmatrix} r_n x(0, \varepsilon) \\ r_n \zeta(T, \varepsilon) \end{pmatrix} + \overset{(i)}{\mathcal{B}}(t)\begin{pmatrix} r_n y(0, \varepsilon) \\ r_n \eta(T, \varepsilon) \end{pmatrix} + \overset{(i)}{\mathcal{C}}(t)\begin{pmatrix} r_n z(0, \varepsilon) \\ r_n \theta(T, \varepsilon) \end{pmatrix}, \ i = \overline{1, 4}.$$

Taking into account the algorithm of the asymptotics construction and the form of the functions $\overset{(i)}{\chi}$, $i = \overline{1, 4}$, we obtain two important properties, namely:

(1) for $t \in [0, T]$, $0 < \varepsilon \leqslant \varepsilon_0$, the following inequalities take place

$$\|\overset{(i)}{\chi}(0, t, \varepsilon)\| \leq c\varepsilon^{n+1}, \ i = 1, 4, \ \|\overset{(3)}{\chi}(0, t, \varepsilon)\| \leq c(\varepsilon^{n+1} + \varepsilon^n \exp(-æt/\varepsilon^2)$$

$$+\varepsilon^n \exp(æ(t - T)/\varepsilon^2)), \ \|\overset{(2)}{\chi}(0, t, \varepsilon)\| \leq c(\varepsilon^{n+1} + \varepsilon^n \exp(-æt/\varepsilon) \tag{61}$$

$$+\varepsilon^n \exp(æ(t - T)/\varepsilon) + \varepsilon^{n-1} \exp(-æt/\varepsilon^2) + \varepsilon^{n-1} \exp(æ(t - T)/\varepsilon^2)),$$

where $c$ and $æ$ are positive constants independent of $t$, $\varepsilon$,

(2) for any $q > 0$, there exist such constants $\delta = \delta(q)$ and $\varepsilon_0 = \varepsilon_0(q)$ that, for $\|v_i\|_{C_{[0,T]}} \leqslant \delta$, $i = 1, 2$, $0 < \varepsilon \leqslant \varepsilon_0$

$$\|\overset{(i)}{\chi}(v_1, t, \varepsilon) - \overset{(i)}{\chi}(v_2, t, \varepsilon)\|_{C_{[0,T]}} \leqslant q\|v_1 - v_2\|_{C_{[0,T]}}, \ i = \overline{1, 4}. \tag{62}$$

It follows from the form of the matrix $\overset{(4)}{\mathcal{C}}(t)$ that the boundary value problem

$$\varepsilon^2\frac{dZ}{dt} = \overset{(4)}{\mathcal{C}}(t)Z, \ Z = (Z_1', Z_2')', \ Z_1(0) = 0, \ Z_2(T) = 0, \tag{63}$$

is uniquely solvable [30]. Therefore, there exists a matrix Green function $\overset{(4)}{G}(t, s, \varepsilon)$ for this problem.

For eigenvalues of the matrix $\overset{(4)}{\mathcal{C}}(t)$, we suppose the condition:

I. $\lambda_i(t) \neq \lambda_j(t)$ for $i \neq j$, $t \in [0, T]$.

Then, in the matrix $\mathfrak{B} = \begin{pmatrix} \mathfrak{B}_{11} & \mathfrak{B}_{12} \\ \mathfrak{B}_{21} & \mathfrak{B}_{22} \end{pmatrix}$, consisting of eigenvectors of the matrix $\overset{(4)}{\mathcal{C}}(t)$, the matrices $\mathfrak{B}_{ii}$, $i = 1, 2$, are nondegenerate. Hence, the condition $4^0$ from ([28], c.125) is valid and therefore due to ([28], n. 9) for sufficiently small $\varepsilon > 0$ the matrix Green function $\overset{(4)}{G}(t, s, \varepsilon)$ satisfies the inequality

$$\|\overset{(4)}{G}(t, s, \varepsilon)\| \leq c\exp(-æ|t - s|/\varepsilon^2), \ t, s \in [0, T]. \tag{64}$$

Furthermore, we need the following three lemmas.

**Lemma 1.** *If $G(t,s)$ is a matrix Green function of the boundary value problem*

$$\frac{dx}{dt} = A(t)x + f(t), t \in [0,T], \ Px(0) = 0, \ (I-P)x(T) = 0,$$

*where the matrix $A(t)$ is continuous with respect to $t$ and invertible for all $t \in [0,T]$, and $P$ is a projector, then*

$$\frac{\partial G(t,s)}{\partial t} = -A(t)\frac{\partial G(t,s)}{\partial s}A(s)^{-1}, \ t \neq s.$$

The proof of this lemma is given in [24]. It follows from the explicit form for the matrix Green function

$$G(t,s) = \begin{cases} -V(t,0)((I-P)V(T,0)(I-P))^{-1}(I-P)V(T,s), \ t \leqslant s, \\ V(t,s) - V(t,0)((I-P)V(T,0)(I-P))^{-1}(I-P)V(T,s), \ t \geqslant s, \end{cases} \tag{65}$$

where $V(t,s) = V(t)V(s)^{-1}$, $dV(t,s)/dt = A(t)V(t,s)$, $V(s,s) = I$.

**Lemma 2.** *The boundary value problem*

$$\mathcal{E}(0)\frac{d\overline{w}}{dt} = A(t)\overline{w} + S(t)\overline{\varphi},$$

$$\mathcal{E}(0)\frac{d\overline{\varphi}}{dt} = W(t)\overline{w} - A(t)'\overline{\varphi},$$

$$\mathcal{E}(0)\overline{w}(0) = 0, \ \mathcal{E}(0)\overline{\varphi}(T) = 0,$$

*where $\overline{w} = (\overline{x}', \overline{y}', \overline{z}')'$, $\overline{\varphi} = (\overline{\xi}', \overline{\eta}', \overline{\theta}')'$, is uniquely solvable.*

**Proof.** Multiply scalarly the first equation of the considered system by $\overline{\varphi}$ and the second equation in this system by $\overline{x}$. Adding the obtained results, we have $d/dt(\overline{\varphi}'\mathcal{E}(0)\overline{w}) = \overline{\varphi}'S(t)\overline{\varphi} + \overline{w}'W(t)\overline{w}$. Integrating this equality over the interval $[0,T]$, in view of the boundary values, we obtain $\int_0^T (\overline{\varphi}'S(t)\overline{\varphi} + \overline{w}'W(t)\overline{w}) \, dt = 0$. Taking into account the positive definiteness of $S(t)$ and $W(t)$, we obtain $\overline{w}(t) = \overline{\varphi}(t) = 0$, i.e., the unique solvability is proved. $\square$

**Lemma 3.** *If $\mathcal{G}$ is a contractive mapping in a Banach space X, $x_0 = 0$, $x_k = \mathcal{G}(x_{k-1})$, $k = 1, 2, ...,$ and $\|x_1\| \leqslant a$, then $\|x_k\| \leqslant a/(1-q)$.*

See the proof of this lemma in [24].

**Theorem 2.** *Solution $\vartheta_*(t,\varepsilon)$ of problem $P_\varepsilon$ for sufficiently small $\varepsilon > 0$, $t \in [0,T]$, satisfy the inequality*

$$\|\vartheta_*(t,\varepsilon) - \widetilde{\vartheta}_n(t,\varepsilon)\| \leq c\varepsilon^{n+1}.$$

**Proof.** The proof of this theorem is based on transforming systems (56)–(59) with boundary values (60) to a system of integral equations, using estimates for matrix Green functions and applying to the obtained system the principle of contractive mappings.

Using Green function $\overset{(4)}{G}(t,s,\varepsilon)$, we have from (59) the integral equation

$$\rho_n Z(t,\varepsilon) = \frac{1}{\varepsilon^2} \int_0^T \overset{(4)}{G}(t,s,\varepsilon)(\overset{(4)}{\mathcal{A}}(s)\rho_n X + \overset{(4)}{\mathcal{B}}(s)\rho_n Y) \, ds + \overset{(4)}{\mathcal{G}}(\rho_n\vartheta,t,\varepsilon), \tag{66}$$

where $\overset{(4)}{\mathcal{G}}(\rho_n v, t, \varepsilon) = 1/\varepsilon^2 \int_0^T \overset{(4)}{G}(t, s, \varepsilon) \overset{(4)}{\chi}(\rho_n v, s, \varepsilon)\, ds$.

In view of (61), (62), and (64), the function $\overset{(4)}{\mathcal{G}}(\rho_n \vartheta, t, \varepsilon)$ satisfies the properties (2) and (3) $\overset{(4)}{\mathcal{G}}(0, t, \varepsilon) \leqslant c\varepsilon^{n+1}$.

Furthermore, we will denote functions, appearing under transformations of the problems (56)–(60) with the properties (2) and (3), by $\overset{(j)}{\mathcal{G}}(\vartheta, t, \varepsilon)$, $j = \overline{1, 4}$. Specific forms of these functions are omitted since they are insignificant for the proof.

In transforming (66), we will use the formula following from (63) and Lemma 1

$$\overset{(4)}{G}(t, s, \varepsilon) = -\varepsilon^2 \frac{\partial}{\partial s}\left(\overset{(4)}{G}(t, s, \varepsilon) \overset{(4)}{\mathcal{C}}(s)^{-1}\right) + \varepsilon^2 \overset{(4)}{G}(t, s, \varepsilon) \frac{d}{ds}\left(\overset{(4)}{\mathcal{C}}(s)^{-1}\right), \ t \neq s. \tag{67}$$

We present the integral in (66), containing the first term on the right side in (67) as the sum of integrals over the intervals $[0, t]$ and $[t, T]$ and integrate by parts. Taking into account the jump of function $\overset{(4)}{G}(t, s, \varepsilon)$ at $s = t$, i.e., the equality $\overset{(4)}{G}(t, t+0, \varepsilon) - \overset{(4)}{G}(t, t-0, \varepsilon) \equiv -I_{2n_3}$, following from (65), and estimate (64), we obtain

$$\rho_n Z(t, \varepsilon) = -\overset{(4)}{\mathcal{C}}(t)^{-1}\left(\overset{(4)}{\mathcal{A}}(t)\rho_n X(t, \varepsilon) + \overset{(4)}{\mathcal{B}}(t)\rho_n Y(t, \varepsilon)\right)$$
$$+ \overset{(4)}{G}(t, 0, \varepsilon) \overset{(4)}{\mathcal{C}}(0)^{-1}\left(\overset{(4)}{\mathcal{A}}(0)\rho_n X(0, \varepsilon) + \overset{(4)}{\mathcal{B}}(0)\rho_n Y(0, \varepsilon)\right) \tag{68}$$
$$- \overset{(4)}{G}(t, T, \varepsilon) \overset{(4)}{\mathcal{C}}(T)^{-1}\left(\overset{(4)}{\mathcal{A}}(T)\rho_n X(T, \varepsilon) + \overset{(4)}{\mathcal{B}}(T)\rho_n Y(T, \varepsilon)\right) + \overset{(4)}{\mathcal{G}}(\rho_n v, t, \varepsilon).$$

Substitute (68) into (58). Introducing the notation $\Lambda(t) = \overset{(3)}{\mathcal{B}}(t) - \overset{(3)}{\mathcal{C}}(t)\overset{(4)}{\mathcal{C}}(t)^{-1}\overset{(4)}{\mathcal{B}}(t)$, we write the obtained equation in the following way:

$$\varepsilon \frac{d\rho_n Y}{dt} = \left(\overset{(3)}{\mathcal{A}}(t) - \overset{(3)}{\mathcal{C}}(t)\overset{(4)}{\mathcal{C}}(t)^{-1}\overset{(4)}{\mathcal{A}}(t)\right)\rho_n X(t, \varepsilon) + \Lambda(t)\rho_n Y(t, \varepsilon)$$
$$+ \overset{(3)}{\mathcal{C}}(t)\left(\overset{(4)}{G}(t, 0, \varepsilon) \overset{(4)}{\mathcal{C}}(0)^{-1}\left(\overset{(4)}{\mathcal{A}}(0)\rho_n X(0, \varepsilon) + \overset{(4)}{\mathcal{B}}(0)\rho_n Y(0, \varepsilon)\right)\right) \tag{69}$$
$$- \overset{(4)}{G}(t, T, \varepsilon) \overset{(4)}{\mathcal{C}}(T)^{-1}\left(\overset{(4)}{\mathcal{A}}(T)\rho_n X(T, \varepsilon) + \overset{(4)}{\mathcal{B}}(T)\rho_n Y(T, \varepsilon)\right)\right) + \overset{(3)}{\mathcal{G}}(\rho_n \vartheta, t, \varepsilon).$$

Let us study the structure of the matrix $\Lambda = \Lambda(t)$.

For brevity, we will sometimes omit the argument $t$. Due to our assumption, it follows from [30] that the Hamiltonian matrix $\overset{(4)}{\mathcal{C}}(t)$ is invertible and its inverse has the form $\overset{(4)}{\mathcal{C}}^{-1} = \begin{pmatrix} D_1 & D_2 \\ D_3 & -D_1' \end{pmatrix}$, where $D_2$ and $D_3$ are symmetric. Similarly to the proof in [30] of the non-negative definiteness of the matrices $D_2$ and $D_3$, it is proved that, in view of the positive definiteness of $S_{33}$ and $W_{33}$, the matrices $D_2$ and $D_3$ are also positive definite.

Let the matrix $\Lambda(t)$ have the block presentation $\begin{pmatrix} \Lambda_1(t) & \Lambda_2(t) \\ \Lambda_3(t) & \Lambda_4(t) \end{pmatrix}$. Write out the explicit expressions for $\Lambda_i(t)$, $i = \overline{1, 4}$:

$$\Lambda_1 = A_{22} - A_{23}(D_1 A_{32} + D_2 W_{23}') - S_{23}(D_3 A_{32} - D_1' W_{23}'),$$
$$\Lambda_2 = S_{22} - A_{23}(D_1 S_{23}' - D_2 A_{23}') - S_{23}(D_3 S_{23}' + D_1' A_{23}'),$$
$$\Lambda_3 = W_{22} - W_{23}(D_1 A_{32} + D_2 W_{23}') + A_{32}'(D_3 A_{32} - D_1' W_{23}'),$$
$$\Lambda_4 = -A_{22}' - W_{23}(D_1 S_{23}' - D_2 A_{23}') + A_{32}'(D_3 S_{23}' + D_1' A_{23}').$$

Comparing $\Lambda_1(t)$ with $\Lambda_4(t)$, it is not difficult to see that $\Lambda_4(t) = -\Lambda_1(t)'$. It also follows from the form of the matrices $\Lambda_2(t)$ and $\Lambda_3(t)$ that these matrices are symmetric.

Introducing for an arbitrary $b \in \mathbb{R}^{n_2}$ the notation

$$b_1 = A'_{23}b, \; b_2 = S'_{23}b, \; b_3 = \begin{pmatrix} b \\ -(D'_1 b_1 + D_3 b_2) \end{pmatrix},$$

$$b_4 = A_{32}b, \; b_5 = W'_{23}b, \; b_6 = \begin{pmatrix} b \\ -(D_1 b_4 + D_2 b_5) \end{pmatrix},$$

we obtain

$$b'\Lambda_2 b = b'_3 \begin{pmatrix} S_{22} & S_{23} \\ S'_{23} & S_{33} \end{pmatrix} b_3 + (D_1 b_2 - D_2 b_1)' W_{33} (D_1 b_2 - D_2 b_1)$$
$$+ b'_1 (D_2 - D_2 W_{33} D_2 - D_1 S_{33} D'_1) b_1 + 2b'_1 (D_2 W_{33} D_1 - D_1 S_{33} D_3) b_2$$
$$+ b'_2 (D_3 - D'_1 W_{33} D_1 - D_3 S_{33} D_3) b_2,$$

$$b'\Lambda_3 b = b'_6 \begin{pmatrix} W_{22} & W_{23} \\ W'_{23} & W_{33} \end{pmatrix} b_6 + (D_3 b_4 - D'_1 b_5)' S_{33} (D_3 b_4 - D'_1 b_5)$$
$$+ b'_4 (D_3 - D_3 S_{33} D_3 - D'_1 W_{33} D_1) b_4 + 2b'_4 (D_3 S_{33} D'_1 - D'_1 W_{33} D_2) b_5$$
$$+ b'_5 (D_2 - D_1 S_{33} D'_1 - D_2 W_{33} D_2) b_5.$$

Taking into account the equalities

$$A_{33}D_1 + S_{33}D_3 = I_{n_3}, \; A_{33}D_2 - S_{33}D'_1 = 0,$$
$$W_{33}D_1 - A'_{33}D_3 = 0, \; W_{33}D_2 + A'_{33}D'_1 = I_{n_3},$$

we obtain that three last summands in the expressions for $b'\Lambda_2 b$ and $b'\Lambda_3 b$ are equal to zero.

In view of positive definiteness of matrices $S(t)$ and $W(t)$, the matrices $\begin{pmatrix} S_{22} & S_{23} \\ S'_{23} & S_{33} \end{pmatrix}$, $S_{33}$, $\begin{pmatrix} W_{22} & W_{23} \\ W'_{23} & S_{33} \end{pmatrix}$, $W_{33}$ are positive definite too. Then, the positive definiteness of matrices $\Lambda_2(t)$ and $\Lambda_3(t)$ follows from the obtained forms for $b'\Lambda_2(t)b$ and $b'\Lambda_3 b$.

Thus, the matrix $\Lambda(t)$ has the form $\begin{pmatrix} \Lambda_1(t) & \Lambda_2(t) \\ \Lambda_3(t) & -\Lambda_1(t)' \end{pmatrix}$, where $\Lambda_2(t)$ and $\Lambda_3(t)$ are positive definite.

We will suppose yet one condition

*II.* Eigenvalues of the matrix $\Lambda(t)$ satisfy the condition *I.*

Then, the boundary value problem

$$\varepsilon \frac{dY}{dt} = \Lambda(t)Y, \; Y = (Y'_1, Y'_2)', \; Y_1(0) = 0, \; Y_2(T) = 0 \tag{70}$$

has a unique solution and, for the corresponding matrix Green function $\overset{(3)}{G}(t,s,\varepsilon)$, the following inequality is valid

$$\| \overset{(3)}{G}(t,s,\varepsilon) \| \leqslant c \exp\left(-\mathit{æ}|t - s|/\varepsilon\right), t, s \in [0, T]. \tag{71}$$

With the help of the Green function $\overset{(3)}{G}(t,s,\varepsilon)$, using (64), (71), we obtain from (69) the following

$$\rho_n Y(t, \varepsilon) = \frac{1}{\varepsilon} \int_0^T \overset{(3)}{G}(t,s,\varepsilon)(\overset{(3)}{\mathcal{A}}(s) - \overset{(3)}{\mathcal{C}}(s)\overset{(4)}{\mathcal{C}}(s)^{-1}\overset{(4)}{\mathcal{A}}(s))\rho_n X(s, \varepsilon)\, ds + \overset{(3)}{\mathcal{G}}(\rho_n \vartheta, t, \varepsilon). \tag{72}$$

The following formula follows from Lemma [1] and (70)

$$\overset{(3)}{G}(t,s,\varepsilon) = -\varepsilon\frac{\partial}{\partial s}\big(\overset{(3)}{G}(t,s,\varepsilon)\Lambda(s)^{-1}\big) + \varepsilon\overset{(3)}{G}(t,s,\varepsilon)\frac{d}{ds}(\Lambda(s)^{-1}), \ t \neq s. \tag{73}$$

Using this formula, we present the integral in (72), containing the first term from the right side (73), as a sum of integrals over the intervals $[0, t]$ and $[t, T]$ and integrate by parts. In view of the jump of function $\overset{(3)}{G}(t,s,\varepsilon)$ at $s = t$ and estimate (71), we have

$$\begin{aligned}
\rho_n Y(t,\varepsilon) &= -\Lambda(t)^{-1}(\overset{(3)}{\mathcal{A}}(t) - \overset{(3)}{\mathcal{C}}(t)\overset{(4)}{\mathcal{C}}(t)^{-1}\overset{(4)}{\mathcal{A}}(t))\rho_n X(t,\varepsilon) \\
&+ \overset{(3)}{G}(t,0,\varepsilon)\Lambda(0)^{-1}(\overset{(3)}{\mathcal{A}}(0) - \overset{(3)}{\mathcal{C}}(0)\overset{(4)}{\mathcal{C}}(0)^{-1}\overset{(4)}{\mathcal{A}}(0))\rho_n X(0,\varepsilon) \\
&- \overset{(3)}{G}(t,T,\varepsilon)\Lambda(T)^{-1}(\overset{(3)}{\mathcal{A}}(T) - \overset{(3)}{\mathcal{C}}(T)\overset{(4)}{\mathcal{C}}(T)^{-1}\overset{(4)}{\mathcal{A}}(T))\rho_n X(T,\varepsilon) + \overset{(3)}{\mathcal{G}}(\rho_n\vartheta,t,\varepsilon).
\end{aligned} \tag{74}$$

Taking into account (68), (74), we obtain from (57) the following equation:

$$\begin{aligned}
\frac{d\rho_n X(t,\varepsilon)}{dt} &= \Omega(t)\rho_n X(t,\varepsilon) + (\overset{(2)}{\mathcal{B}}(t) - \overset{(2)}{\mathcal{C}}(t)\overset{(4)}{\mathcal{C}}(t)^{-1}\overset{(4)}{\mathcal{B}}(t)) \\
&\times(\overset{(3)}{G}(t,0,\varepsilon)\Lambda(0)^{-1}(\overset{(3)}{\mathcal{A}}(0) - \overset{(3)}{\mathcal{C}}(0)\overset{(4)}{\mathcal{C}}(0)^{-1}\overset{(4)}{\mathcal{A}}(0))\rho_n X(0,0) \\
&- \overset{(3)}{G}(t,T,\varepsilon)\Lambda(T)^{-1}(\overset{(3)}{\mathcal{A}}(T) - \overset{(3)}{\mathcal{C}}(T)\overset{(4)}{\mathcal{C}}(T)^{-1}\overset{(4)}{\mathcal{A}}(T))\rho_n X(T,\varepsilon) \\
&+ \overset{(2)}{\mathcal{C}}(t)(\overset{(4)}{G}(t,0,\varepsilon)\overset{(4)}{\mathcal{C}}(0)^{-1}(\overset{(4)}{\mathcal{A}}(0)\rho_n X(0,\varepsilon) + \overset{(4)}{\mathcal{B}}(0)\rho_n Y(0,\varepsilon)) \\
&- \overset{(4)}{G}(t,T,\varepsilon)\overset{(4)}{\mathcal{C}}(T)^{-1}(\overset{(4)}{\mathcal{A}}(T)\rho_n X(T,\varepsilon) + \overset{(4)}{\mathcal{B}}(T)\rho_n Y(T,\varepsilon)) \\
&+ \overset{(2)}{\chi}(\rho_n\vartheta,t,\varepsilon) + \overset{(2)}{\mathcal{G}}(\rho_n\vartheta,t,\varepsilon),
\end{aligned} \tag{75}$$

where

$$\begin{aligned}
\Omega(t) = \overset{(2)}{\mathcal{A}}(t) - \overset{(2)}{\mathcal{C}}(t)\overset{(4)}{\mathcal{C}}(t)^{-1}\overset{(4)}{\mathcal{A}}(t) - (\overset{(2)}{\mathcal{B}}(t) - \overset{(2)}{\mathcal{C}}(t)\overset{(4)}{\mathcal{C}}(t)^{-1}\overset{(4)}{\mathcal{B}}(t))\Lambda(t)^{-1}(\overset{(3)}{\mathcal{A}}(t) \\
- \overset{(3)}{\mathcal{C}}(t)\overset{(4)}{\mathcal{C}}(t)^{-1}\overset{(4)}{\mathcal{A}}(t)).
\end{aligned}$$

It follows from Lemma [2] that the boundary value problem

$$\frac{d}{dt}\begin{pmatrix} \bar{x} \\ \bar{\zeta} \end{pmatrix} = \Omega(t)\begin{pmatrix} \bar{x} \\ \bar{\zeta} \end{pmatrix}, \ \bar{x}(0) = 0, \ \bar{\zeta}(T) = 0$$

is uniquely solvable. Hence, there exists the matrix Green function $\overset{(2)}{G}(t,s)$ of the last boundary value problem, which is bounded, i.e.,

$$\|\overset{(2)}{G}(t,s)\| \leq c, \ t,s \in [0,T]. \tag{76}$$

Due to (61), (62), (64), (71), and (76) from the expression of the solution $\rho_n X(t,\varepsilon)$ of Equation (75) with the boundary values from (60), written by the help of the matrix Green function $\overset{(2)}{G}(t,s)$, it follows that $\rho_n X(t,\varepsilon) = \overset{(2)}{\mathcal{G}}(\rho_n\vartheta,t,\varepsilon)$. Furthermore, from (74), (68), and (56), and properties (1), (2), we successively obtain $\rho_n Y(t,\varepsilon) = \overset{(3)}{\mathcal{G}}(\rho_n\vartheta,t,\varepsilon)$,

$\rho_n Z(t, \varepsilon) = \overset{(4)}{\mathcal{G}}(\rho_n \vartheta, t, \varepsilon)$, $r_n u(t, \varepsilon) = \overset{(1)}{\mathcal{G}}(\rho_n \vartheta, t, \varepsilon)$. Thus, for determining $\rho_n \vartheta$, we have in the space of continuous functions with values in $\mathbb{R}^m \times \mathbb{R}^{n_1} \times \mathbb{R}^{n_2} \times \mathbb{R}^{n_3}$ the equation

$$\rho_n \vartheta = \mathcal{G}(\rho_n \vartheta, t, \varepsilon), \tag{77}$$

where the function $\mathcal{G}$ for sufficiently small $\varepsilon > 0$, $t \in [0, T]$ satisfies properties (2), (3). If we will take in condition (2) $q < 1$, then $\mathcal{G}$ is a contraction mapping in $C_{[0,T]}$. According to the contractive mappings principle, Equation (77) has a unique solution, and this solution can be found by the method of successive approximations.

According to (3) and Lemma 3, all successive approximations are not more than $c\varepsilon^{n+1}$. Hence, a solution of the Equation (77) will have the same estimate. Due to (49) and (55), it proves the theorem statement. $\square$

**Theorem 3.** *Under conditions of Theorem 2, for sufficiently small $\varepsilon > 0$, the following inequality for the performance index is valid*

$$J_\varepsilon(\widetilde{u}_n) - J_\varepsilon(u_*) \leq c\varepsilon^{2n+2}.$$

**Proof.** Denoting by $\widetilde{s}$ a solution of the problem (2)–(3) at $u = \widetilde{u}_n$, we present the solution of the problem $P_\varepsilon$ in the form $w_* = \widetilde{s} + \delta w$, $u_* = \widetilde{u}_n + \Delta u$, then $\delta w$ satisfies the system

$$\mathcal{E}(\varepsilon)\frac{d\delta w}{dt} = A(t)\delta w + B(t)\Delta u + \varepsilon(f(w_*, u_*, t, \varepsilon) - f(w_* - \delta w, u_* - \Delta u, t, \varepsilon)),$$

$$\delta w(0, \varepsilon) = 0.$$

In view of Theorem 2,

$$\|\Delta u\| = \|r_n u\| \leqslant c\varepsilon^{n+1}. \tag{78}$$

Using this estimate and the condition of stability of the matrices $A_{33}$ and $A_{22} - A_{23}A_{33}^{-1}A_{32}$, we can prove the estimate

$$\|\delta w(t, \varepsilon)\| \leqslant c\varepsilon^{n+1}. \tag{79}$$

Introducing the notation $\Delta J = J_\varepsilon(\widetilde{u}_n) - J_\varepsilon(u_*)$, we present it in the form

$$\Delta J = \frac{1}{2}\int_0^T (\delta w' W(t)\delta w + \Delta u' R(t)\Delta u)\, dt$$

$$+ \int_0^T (-\delta w' W(t)w_* - \Delta u' R(t)u_* + \varepsilon(F(\widetilde{w}, \widetilde{u}_n, t, \varepsilon) - F(w_*, u_*, t, \varepsilon))\, dt.$$

Using control optimality condition (27), (28) for the problem (1)–(3), after integrating by parts and taking into account the equation for $\delta w$ and the boundary values $\delta w(0, \varepsilon)$, $\varphi(T, \varepsilon)$, we have

$$\Delta J = \frac{1}{2}\int_0^T (\delta w' W(t)\delta w + \Delta u' R(t)\Delta u)\, dt + \varepsilon \int_0^T (\varphi'(f(w_*, u_*, t, \varepsilon)$$

$$- f(w_* - \delta w, u_* - \Delta u, t, \varepsilon) - f_w(w_*, u_*, t, \varepsilon)\delta w - f_u(w_*, u_*, t, \varepsilon)\Delta u)$$

$$+ F(w_* - \delta w, u_* - \Delta u, t, \varepsilon) - F(w_*, u_*, t, \varepsilon) + F_w(w_*, u_*, t, \varepsilon)\delta w + F_u(w_*, u_*, t, \varepsilon)\Delta u)\, dt.$$

From here, in view of (78) and (79), we obtain the theorem assertion. $\square$

Denote by $\widetilde{u}_{i*}$ the $i$-th order approximation for an optimal control constructed by the direct scheme method.

**Theorem 4.** *For sufficiently small $\varepsilon > 0$, the following inequalities are valid*

$$J_\varepsilon(\widetilde{u}_{*(n-1)}) \geqslant J_\varepsilon(\widetilde{u}_{*(n-1)} + \varepsilon^n \overline{u}_{*n})$$

$$\geqslant J_\varepsilon(\widetilde{u}_{*(n-1)} + \varepsilon^n(\overline{u}_{*n} + \Pi_{0n}u_* + Q_{0n}u_*)) \geqslant J_\varepsilon(\widetilde{u}_{*n}), \ n \geqslant 1. \tag{80}$$

*If an addition to $\widetilde{u}_{*(n-1)}$ is non-zero, then the corresponding inequality is strict.*

**Proof.** The proof of this theorem is based on Theorem 1.

Asymptotic solution of the form (4) can be constructed for a solution of problem (2), (3) at control $u = \widetilde{u}_{*n}$ for each n. Moreover, the terms of these asymptotic solution coincide with corresponding terms of asymptotic expansion of optimal trajectory to *n*-th order inclusively.

Substitute expansions for $\widetilde{u}_{*(n-1)}(t, \varepsilon)$, $\widetilde{u}_{*n}(t, \varepsilon)$ and corresponding trajectories into $J_\varepsilon(u)$. After applying the expansion (6) and Theorem 1, we see that the first $2n$ coefficients from (6) coincided and a difference between $J_\varepsilon(\widetilde{u}_{*(n-1)})$ and $J_\varepsilon(\widetilde{u}_{*n})$ appears for the first time in the coefficient $J_{2n}$.

Taking into account the equality $\widetilde{u}_{*n} = \widetilde{u}_{*(n-1)} + \varepsilon^n(\overline{u}_{*n} + \sum_{i=0}^{1}(\Pi_{in}u_* + Q_{in}u_*))$, consider the expressions for the coefficients $J_{2n}$, $J_{2n+1}$ and $J_{2n+2}$ separately.

If we omit the terms known after solving problems $\overline{P}_j$, $\Pi_{0j}P$, $Q_{0j}P$, $j = \overline{0, n-1}$, $\Pi_{1j}P$, $Q_{1j}P$, $j = \overline{0, n-2}$, we obtain from $J_{2n}$ the sum of the performance indices $\overline{J}_n + \Pi_{1(n-1)}J + Q_{1(n-1)}J$. For $J_\varepsilon(\widetilde{u}_{*n})$, we have $\overline{J}_n = \overline{J}_n(\overline{u}_n)$ and, for $J_\varepsilon(\widetilde{u}_{*(n-1)})$, we have $\overline{J}_n = \overline{J}_n(0)$. Since $\overline{u}_{*n}$ is found from the problem of minimizing the functional $\overline{J}_n$, we obtain the first inequality in (80).

By a similar way, we obtain from the form $J_{2n+1}$ the sum $\Pi_{0n}J + Q_{0n}J$. For $J_\varepsilon(\widetilde{u}_{*n})$, we have $\Pi_{0n}J = \Pi_{0n}J(\Pi_{0n}u_*)$, $Q_{0n}J = Q_{0n}J(Q_{0n}u_*)$. For $J_\varepsilon(\widetilde{u}_{*(n-1)})$, we have $\Pi_{0n}J = \Pi_{0n}J(0)$, $Q_{0n}J = Q_{0n}J(0)$. Since $\Pi_{0n}u_*$ and $Q_{0n}u_*$ are found by means of minimizing the functionals $\Pi_{0n}J$ and $Q_{0n}J$, respectively, we obtain the second inequality in (80).

Analogously, the third inequality in (80) follows from the form $J_{2n+2}$.

The assertion concerning non-zero additions to $\widetilde{u}_{*(n-1)}$ follows from the unique solvability of linear-quadratic control problems. $\square$

**Remark 3.** *From Theorems 3 and 4, it follows that the sequences $\{\widetilde{u}_{*(n-1)}\}$, $\{\widetilde{u}_{*(n-1)} + \varepsilon^n \overline{u}_{*n}\}$, $\{\widetilde{u}_{*(n-1)} + \varepsilon^n(\overline{u}_{*n} + \Pi_{0n}u_* + Q_{0n}u_*)\}$ are minimizing.*

## 6. Conclusions

In this paper, unlike the previous one [26], devoted to a similar problem, detailed proofs of linear-quadratic optimal control problems forms, from which terms of asymptotic solution of given nonlinear optimal control problem are found, are presented in Theorem 1. Note that all problems for finding asymptotic terms are obtained in an explicit form. It is very comfortable for research applying asymptotic methods for solving practical problems.

For the first time, asymptotic estimates of the proximity between the exact and asymptotic solutions are established for the control, state trajectory in Theorem 2 and for the minimized functional in Theorem 3.

It should be noted that, in view of Theorem 4, values of the minimized functional with a control, which is an asymptotic approximation to the optimal control $u_*$ respectively of the form $\widetilde{u}_{*(n-1)}$, $\widetilde{u}_{*(n-1)} + \varepsilon^n \overline{u}_{*n}$, $\widetilde{u}_{*(n-1)} + \varepsilon^n(\overline{u}_{*n} + \Pi_{0n}u_* + Q_{0n}u_*)$, $\widetilde{u}_{*n}$, do not increase. It follows from Theorem 3 that the corresponding sequences of the controls are minimizing.

In the future, it is useful to give a program realization of applying the direct scheme method for problems of type (1)–(3). The results obtained in the paper can be used for constructing asymptotic solutions of practical optimal control problems with three-tempo state variables and weak nonlinear perturbations in a linear state equation and a quadratic performance index.

The advantage of applying a direct scheme method is the possibility to use standard software packages for solving optimal control problems in order to find terms of asymptotic

solution. As it is proved in this paper, for problems with three-tempo state variables, the found sequence of approximations to the optimal control $\{\widetilde{u}_{*n}\}$ is minimizing.

**Author Contributions:** Problem statement and methodology, G.K.; algorithm of asymptotics construction, G.K. and M.K.; asymptotic estimates, G.K.; writing—original draft preparation, M.K.; writing—review and editing G.K. All authors have read and agreed to the published version of the manuscript.

**Funding:** The work of the first author was supported by the Russian Science Foundation (Project No. 21-11-00202).

**Institutional Review Board Statement:** Not applicable.

**Informed Consent Statement:** Not applicable.

**Data Availability Statement:** Not applicable.

**Acknowledgments:** The authors thank the three Reviewers for helpful comments and N.T. Hoai and A.S. Kostenko for useful discussions.

**Conflicts of Interest:** The authors declare no conflict of interest. The funders had no role in the design of the study; in the collection, analysis, or interpretation of data; in the writing of the manuscript, or in the decision to publish the results.

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
