# Peer review of "Justification of Direct Scheme for Asymptotic Solving Three-Tempo Linear-Quadratic Control Problems under Weak Nonlinear Perturbations"

_axioms, doi:10.3390/axioms11110647_

Round 1

Reviewer 1 Report

The writing in general is acceptable.The article contains some novel part. The topic addressed in the paper is potentially interesting and the results presented in this paper seem correct. 

1) To highlight the contribution, some sentences should be added in introduction section and its can be itemized.

2) The authors should refer any further developments about the proposed approach and also refer any application of this strategy. Both topics should be discussion in the conclusion section.

3) The disadvantage or the limitation of the proposed method must be described in conclusion

4) All assumptions and constraints should be discussed.

5) In the simulation part, more discussions are expected to interpret the simulation results obtained. How do you ensure the results are enough to verify the proposal? And how the parameters set?  How do you ensure the comparisons be fair?

Reviewer 2 Report

The paper has merits and very well written, I am impressed by the work presented in the draft . Please go through some minor typos and English grammar correction. 

Reviewer 3 Report

In this work, an asymptotics and approximated solution in the framework of weakly nonlinearly perturbed linear-quadratic optimal control problem with three-tempo state variables is constructed. Also, the estimates of the proximity between the asymptotic and exact solutions are proved for the control, state trajectory and minimized functional. Nonincreasing of the minimized functional, if a next approximation to the optimal control is used, follows from the proposed algorithm of the asymptotics construction. The validation of solution of the obtained algorithm is testing by  applied it to a dynamical system of differential equations with comparison to the exact solution   

In conclusion: Some results obtained in this work are novel. Moreover, they are incremental improvements of earlier results. The level of mathematical and physical difficulty and originality of these results makes them suitable for publication. The experts in this field of the perturbation theory of differential equations will appreciate some technical progress exposed in this work. However, some minor comments

·      The abstract must be rewritten to show clearly the new results.

·      However, the introduction is good, but I think that it can be improved

·      The resolution of diagrams must be improved with using code color

·      The author have to investigate the similarities and differences among their work and the literatures of this proposed work 

·      The conclusion section must be added

In general, the authors have to describe in more detail the purpose of their study and its original contents. A clear explanation, what is the new result in their work, and how it is build up upon previous work in the field.

If the authors submit a modified version according to my suggestions where they also give more details/explanations about the abovementioned criticisms, I could recommend the paper for publication.
